# `RandomFront 2.3` **A physical parametrisation of fire-spotting for operational fire spread models: Implementation in `WRF-Sfire` and response analysis with `LSFire+`**

Andrea Trucchia[1,2], Vera Egorova[1], Anton Butenko[3,4], Inderpreet Kaur[5], and Gianni Pagnini[1,6]

[1]BCAM–Basque Center for Applied Mathematics, Bilbao, Basque Country – Spain
[2]Department of Mathematics, University of the Basque Country UPV/EHU, Bilbao, Basque Country – Spain
[3]Space Research Institute of Russian Academy of Sciences, Moscow, Russia
[4]Institute of Geography, University of Bremen, Bremen, Germany
[5]Department of Atmospheric Chemistry, Max Planck Institute for Chemistry, Mainz, Germany
[6]Ikerbasque–Basque Foundation for Science, Bilbao, Basque Country – Spain

**Correspondence:** Gianni Pagnini (gpagnini@bcamath.org)

**Abstract.** Fire-spotting is often responsible for a dangerous flare up in the wildfire and causes secondary ignitions isolated from the primary fire zone leading to perilous situations. The main aim of the present research to provide a versatile probabilistic model for fire-spotting that is suitable for implementation as a post-processing scheme at each time step in any of the existing operational large-scale wildfire propagation models, without calling for any major changes in the original framework. In particular, a complete physical parametrisation of fire-spotting is presented and the corresponding updated model `RandomFront 2.3` is implemented in a coupled fire-atmosphere model : `WRF-Sfire`. A test case has been simulated and discussed. Moreover, the results from different simulations with a simple model based on the Level Set Method, namely `LSFire+`, highlight the response of the parametrisation to varying fire intensities, wind conditions and different firebrand radii. The contribution of the firebrands towards increasing the fire perimeter varies according to different concurrent conditions and the simulations show results in agreement with the physical processes. Among the many rigorous approaches available in literature to model the firebrand transport and distribution, the approach presented here proves to be simple yet versatile for application to operational large-scale fire spread models.

## 1 Introduction

Fire-spotting is an important phenomenon associated with the wildfires (Fernandez-Pello, 2017). It is documented as a dominant phenomenon contributing towards a rampant spread of fire in many devastating historical fires (Koo et al., 2010). Spot fires occur when fragments of the fuel tear off from the main fuel source and the horizontal wind transports the burning embers

beyond the zone of direct ignition. The burning embers/firebrands can develop new secondary ignition spots and lead to a perilous increase in the effective rate of spread (ROS) of the fire.

The main aim of the present research is to provide a versatile probabilistic model for fire-spotting that is suitable for implementation as a post-processing scheme at each time step in any of the existing large-scale operational codes for simulating wildfire propagation, without calling for any major changes in the original framework.

Researchers have tried to understand the phenomenology of fire-spotting through both experimental and theoretical aspects to update the existing wildfire management decision support systems. Most of the experimental procedures for studying the fire-spotting phenomenon focus on characterization of the generation of firebrands (Manzello et al., 2007; El Houssami et al., 2016; Thomas et al., 2017), shape and size of the firebrands (Manzello et al., 2009; Tohidi et al., 2015), drag forces and ignition processes (Manzello et al., 2008). The short temporal and spatial scales of the experiments limit a detailed description of the landing distributions and the flight paths of the firebrands. On the other hand, the firebrand transport models provide an estimate of the maximum landing distance and flight paths of the firebrands through a simplified overview of the physical dynamics of the fire behavior, plume characteristics, and the atmospheric conditions around the fire. Tarifa et al. (1965, 1967) and Albini (1979, 1983) were the foremost to develop simplified plume models for an estimation of firebrand lifetimes, flight paths and the potential fire-spotting distance. Beginning with their works, there has been a paradigm shift in the development of the firebrand transport models, with the latest models benefiting from the advanced computational techniques and resources.

Woycheese et al. (1999) provide a model for the lofting of spherical and cylindrical firebrands by using the plume model proposed by Baum and McCaffrey (1989). They suggest analytical functions for the maximum loftable diameter and the maximum loftable height in terms of the fire intensity, atmospheric wind and the fuel characteristics. Numerical experiments by Sardoy and co-workers (Sardoy et al., 2007, 2008) also analyze the effect of atmospheric conditions, fire properties and fuel properties on the firebrand behavior and provide a statistical estimate of the ground level distributions of the disk shaped firebrands. Their results highlight that firebrands landing at short distances (up to $1000\,\text{m}$ from the source) follow a lognormal distribution. A study by Wang (2011) also provides a mathematical model to quantify the distribution and the mass of the firebrands through a Rayleigh distribution function. In an another study, Koo et al. (2007) present a physics based multiphase transport model for wildfires (`FIRETEC`) to study the firebrand transport. In a recent study, Martin and Hillen (2016) also discuss the underlying physical processes for firebrands in detail and they derive a landing distribution based on these physical processes. Besides these statistical approaches, few numerical models based on Large Eddy Simulation (LES) (Himoto and Tanaka, 2005; Pereira et al., 2015; Thurston et al., 2017; Tohidi and Kaye, 2017) or Computational Fluid Dynamics (CFD) (Wadhwani et al., 2017), small world networks (Porterie et al., 2007), cellular automata models (Perryman et al., 2013) also exist in the literature. Bhutia et al. (2010) present one such study based on coupled fire/atmosphere LES for predicting the short range fire-spotting. They simulate multiple firebrand trajectories for analyzing the sensitivity of the flight path to different particle sizes, release heights and wind conditions but also mention the limited applicability of such models to operational use due to the computational demands.

Despite the presence of multiple studies focusing on the detailed aspects of the firebrand landing distributions, none of them is able to provide a comprehensive yet versatile approach for an application to operational fire spread models. The continuing

demand for the operational management tools is to provide a quick and efficient output with simple inputs but at the same time taking the most important parameters into consideration. Few operational fire spread models like `FARSITE` (Finney, 1998), `BEHAVEPLUS` (Andrews and Chase, 1989) and `Prometheus` (Tymstra et al., 2010) incorporate the phenomenon of fire-spotting through the Albini's model (Albini, 1979, 1983). But Albini's model provides only an estimate of the maximum

distance for a spot fire and does not include any function for the ignition probability to model the spread of spot fires. The Australian wildfire simulator `PHOENIX Rapidfire` (Tolhurst et al., 2008) is designed to model large fast moving fires and also includes a fire-spotting module, but the formulations for fire spread in `PHOENIX` are calibrated for eucalyptus forests and a generic application to other types of fuels requires a re-calibration (Pugnet et al., 2013). The new operational models like `WRF-Sfire` (Mandel et al., 2011) and `FOREFIRE` (Filippi et al., 2009) are fast and allow coupling with the atmospheric

models for a better representation of the initial and concurrent atmospheric conditions; but lack any specific module to tackle the fire-spotting behavior.

In this article, the authors proceed with the statistical formulation `RandomFront` for including the effects of random processes into wildfire simulators, namely turbulence and fire-spotting phenomena. The chronology of this approach refers to the following papers: v1.0 includes only turbulence, with no parametrisation (Pagnini and Massidda, 2012a, b), v2.0 includes

turbulence and fire-spotting with literature parametrisation for fire-spotting (Pagnini and Mentrelli, 2014), v2.1 includes turbulence and fire-spotting with parametrisation for turbulence (Kaur et al., 2016) and v2.2 includes turbulence and fire-spotting with a first physical parametrisation of fire-spotting (Kaur and Pagnini, 2016). Finally, in the present version v2.3 the parametrisation of fire-spotting has been modified and corrected (also in view of a remark by one of the Referees) with respect to the previous version.

The physical parametrisation of the probabilistic model is developed to incorporate the fire-spotting behaviour in terms of the fire intensity, wind conditions and fuel characteristics. This formulation is independent of the method used for the fire-line propagation and the definition of the ROS, and it is versatile enough to be utilized with any of the existing operational fire spread models. In their previous work (Kaur et al., 2016), the authors demonstrate the applicability of the formulation to two wildfire models based on different fire-line propagation methods, i.e., a Eulerian moving interface method based on the Level

Set Method (LSM) that is the basis for the `WRF-Sfire` model and a Lagrangian front tracking technique based on the Discrete Event System Specification (DEVS) that is the basis for the `FOREFIRE` model. The aim of the present study is to provide a simple yet complete addition to operational fire spread models for representing the random behaviour of fire-spotting through simple inputs related to the wildfires. This probabilistic model is devised to provide a physical meaning to the spread of fire by virtue of firebrands. The proposed parametrisation has been implemented into `WRF-Sfire` (Coen et al., 2012; Mandel et al.,

2014) and a paradigmatic test case has been simulated. Moreover, the proposed formulation for including random process and the corresponding physical parametrisation are implemented into a much simpler fire spread simulator, also based on the LSM, namely `LSFire+` (Pagnini and Massidda, 2012a, b; Pagnini and Mentrelli, 2014; Chu and Prodanović, 2009; Bevins, 1996). Results from different test cases with `LSFire+` are presented to highlight the sensitivity of the simple parametrisation in simulating the generation of secondary fires by fire-spotting under different wind conditions, fire intensities and firebrand

radii.

The article is organised as follows. In Section 2 a very brief description of the mathematical model is presented, while the physical parametrisation of fire-spotting within the framework of a lognormal distribution of the landing distance is described in Section 3. The implementation of `RandomFront 2.3` in the computational environments `WRF-Sfire` together with a test case and the corresponding discussion are reported in Section 4, and the implementation in `LSFire+` and the corresponding response analysis are discussed in Section 5. The conclusions are drawn in Section 6.

## 2  Model formulation

A mathematical model to represent the random effects associated with the wildland fires has been developed by Pagnini and co-authors (Pagnini and Massidda, 2012b, a; Pagnini, 2013, 2014; Pagnini and Mentrelli, 2016, 2014; Kaur et al., 2015, 2016; Mentrelli and Pagnini, 2016). This formulation describes the motion of the fire-line as a composition of the drifting part and the fluctuating part. The drifting part represents the fire-perimeter obtained through the definition of the ROS based on fuel characteristics and the averaged fire properties. The output from most of the existing operational fire spread models can be considered as the drifting part. On the other hand, the fluctuating part is independent of the drifting part and represents the additional contribution to the fire-perimeter as an effect of the random processes like turbulence and fire-spotting. This model can be implemented as a crucial addition to operational fire spread models through a post processing application at each time step. The drifting component obtained from the output of any wildfire model can be updated with the fluctuating component at each time step to include the effects of turbulence and fire-spotting. A brief overview of the mathematical details has been provided in this section; for a detailed description the interested readers are referred to (Pagnini and Mentrelli, 2014; Kaur et al., 2016).

In a domain $S$, let $\Omega \subseteq S$ represent the burnt area and let $X^\omega = X + \eta^\omega$ represent the trajectory of each active fire point as the sum of a drifting part $X$ and a fluctuating part $\eta^\omega$. The drifting part $X$ is obtained from the output of a wildfire propagation model, while the fluctuations in the fire-line are included through a probability density function (PDF) corresponding to the type of random process under consideration. Let the area enclosed by the drifting part be described through an indicator function $I_\Omega(\boldsymbol{x}, t) = 1$ when $\boldsymbol{x}$ is inside the domain $\Omega$, and $I_\Omega(\boldsymbol{x}, t) = 0$ when $\boldsymbol{x}$ is outside. Considering the ensemble average of the active burning points, a new effective indicator function is defined as:

$$\phi_e(\boldsymbol{x}, t) = \int_S I_\Omega(\overline{\boldsymbol{x}}, t) f(\boldsymbol{x}; t | \overline{\boldsymbol{x}}) \, d\overline{\boldsymbol{x}}, \tag{1}$$

where, $f(\boldsymbol{x}; t | \overline{\boldsymbol{x}})$ represents the PDF which accounts for the fluctuations of the random effects. The effective indicator $\phi_e \in [0, 1]$ and an arbitrary threshold is fixed to mark points as burned, i.e., $\Omega_e(\boldsymbol{x}, t) = \{\boldsymbol{x} \in S \mid \phi_e(\boldsymbol{x}, t) > \phi_e^{th}\}$. The ignition of the fuel by the firebrands involves heat exchange over a sufficient period of time, hence, a sufficient delay is also incorporated in the model through an other function $\psi$. The function $\psi$ simulates the ignition of fuel by hot air and burning embers as an accumulative process over time (heating-before-burning mechanism):

$$\psi(\boldsymbol{x}, t) = \int_0^t \phi_e(\boldsymbol{x}, \eta) \frac{d\eta}{\tau}, \tag{2}$$

where $\tau$ is the ignition delay. At each time $t$, all points $\boldsymbol{x}$ that satisfy the conditions $\psi(\boldsymbol{x}, t) > 1$ or $\phi_e(\boldsymbol{x}, t) > \phi_e^{th}$ are labelled as burned.

The shape of the PDF $f(\boldsymbol{x}; t | \overline{\boldsymbol{x}})$ is established by analyzing the random processes under consideration. The diversity in the shapes of the PDF provides the model a multifaceted outlook. Assuming fire-spotting to be a downwind phenomenon occurring in turbulent atmosphere, the shape of the PDF is defined as follows:

$$
f(\boldsymbol{x}; t | \overline{\boldsymbol{x}}) = \begin{cases} \displaystyle\int\limits_0^\infty G(\boldsymbol{x} - \overline{\boldsymbol{x}} - l\,\hat{\boldsymbol{n}}_U; t) q(l)\, dl, & \text{downwind}, \\[2em] G(\boldsymbol{x} - \overline{\boldsymbol{x}}; t), & \text{upwind}. \end{cases} \tag{3}
$$

The distribution function $G(\boldsymbol{x} - \overline{\boldsymbol{x}}; t)$ is an isotropic bi-variate Gaussian and provides for the effect of the turbulent heat fluxes in fire propagation while, the distribution function $q(l)$ represents the firebrand landing distribution. The strength of the turbulence around the fire is parametrised through a turbulent diffusion coefficient $\mathscr{D}$. A short description of the physical characterisation of $\mathscr{D}$ is presented in the next Section. A precise description of the landing distributions through experimental observations is difficult due to temporal and spatial constraints. But the experimental results analysing the flight paths, shape and landing distributions of the firebrands have shown that the frequency of the firebrands landing in the positive direction from the source increases with distance to a maximum value and then gradually decays to zero (Hage, 1961). The landing distributions of the firebrands have also been studied though the numerical solution of the energy balance equations (Sardoy et al., 2008; Himoto and Tanaka, 2005; Kortas et al., 2009). Among the different transport models proposed in literature, both Sardoy et al. (2008) and Himoto and Tanaka (2005) describe the lognormal density function as an approximate fit to the landing distribution of the firebrands. Whereas, Wang (2011) proposes a Rayleigh distribution for the same. In this article, the shape of $q(l)$ is defined by a lognormal distribution to describe the frequency profile of the fallen firebrands:

$$
q(l) = \frac{1}{\sqrt{2\pi}\sigma l} \exp\frac{-(\ln l/\mu)^2}{2\sigma^2}, \tag{4}
$$

where $\mu$ is the ratio between the square of the mean of landing distance $l$ and its standard deviation, while $\sigma$ is the standard deviation of $\ln l/\mu$.

## 3 Physical parametrisation of fire-spotting

The firebrands generated from the vegetation face strong buoyant forces and the ones with size less than the maximum loftable size are uplifted vertically in the convective column. These firebrands rise to a maximum height till the buoyant and the gravitational forces counterbalance each other. Once the firebrands are expelled from the column, they are steered by the atmospheric wind and they fly at their terminal velocity of fall. The simplified models for the landing distance assume that the ejection of the firebrands from the vertical convective column is a random process affected by the turbulence in the environment around the fire. Among other factors, the strength of the convective column, the atmospheric conditions and the dimensions

of the firebrands play a vital role in governing the trajectory of the firebrands. In this section, the landing distribution of the firebrands based on a lognormal probability function is combined with the physical characterization of the firebrand transport. The parametrisation presented here is simplified and includes only the vital ingredients necessary to describe the firebrand transport. Each firebrand is assumed to be spherical and for a particular set of concurrent atmospheric conditions and fuel characteristics the size is assumed to be constant. Any modification in the flight of the firebrand due to rotation of firebrand or collision with other firebrands is also neglected. Preliminary results were discussed in Kaur and Pagnini (2016).

Literature studies identify the maximum spotting distance as a numerical measure to assess the severity of the fire-spotting danger under different circumstances (Albini, 1979; Tarifa et al., 1965, 1967). Recognizing the importance of maximum-spotting distance, we select to parametrize the mathematical model in terms of the $p^{th}$ percentile of a lognormal distribution as a measure of the maximum landing distance. The $p^{th}$ percentile for a lognormal distribution is described by its location and shape parameters $\mu$ and $\sigma$ respectively:

$$\mathcal{L} = \mu \exp(z_p \sigma), \tag{5}$$

where the value of $z_p$ corresponding to the $p^{th}$ percentile can be estimated from the z-tables (see, for example, `http://www.itl.nist.`). We hypothesize that the $p^{th}$ percentile represents the maximum landing distance for firebrands under different situations and no ignition is possible beyond this cut-off. To ascertain the value of "cut-off" percentile, it is assumed that the effective contribution of the firebrands ceases to be meaningful when its probability reduces to 20 times its peak value. Thereafter, the ability of the firebrands to cause an ignition is assumed to be negligible. The cut-off criteria is chosen empirically, but a sufficiently large number (like 20) ensures that we do not miss out on any considerable fire-spotting behavior existing outside this range.

For this particular distribution, the cut-off for 50th percentile lies way beyond the point denoting the 1/20th of the maximum probability. In order to define a generalized value of the cut-off percentile for all the simulation cases presented in this article, the value of $z_p$ is chosen to be 0.45, which corresponds to the $67^{th}$ percentile point.

The process of fire-spotting can be roughly segregated into generation, lofting and transport of firebrands. The generation of firebrands from a burning canopy is a random and dynamic process; while the lofting and transport of the firebrands is regulated by the firebrand geometry, fuel combustion rates, plume dynamics and ambient wind conditions. The firebrands generated from the vegetation face strong buoyant forces and the ones with size less than the maximum loftable size are uplifted vertically in the convective column. These firebrands rise to a maximum height till the buoyant and the gravitational forces counterbalance each other. Once the firebrands are expelled from the column, they are steered by the atmospheric wind and they fly at their terminal velocity of fall. The different sub-process involved in the firebrand lofting and transport interact with each other and affect the maximum spotting distances. An explicit modeling of the coupled processes is difficult and often different approximations and assumptions are used to simplify the physical processes. One such important works on fire-spotting distributions and maximum spotting distances are by Tarifa and co-workers (Tarifa et al., 1965, 1967). In their different works they describe the spotting distributions and the maximum spotting distances by combining a series of experimental and theoretical approaches. We follow these existing approaches and formulate the physical parametrization for our model. Below we provide a brief discussion of the different processes which are considered in the physical parametrization:

1. Firebrand lofting

   (a) Vertical gas flow: In a convective column, the updraft introduced by fire lifts the firebrands in the convective column. The strength of vertical gas flow $U_{\text{gas}}$ increases with the fire intensity $I$ and is empirically expressed as in (Muraszew, 1974):

   $$U_{\text{gas}} = 9.35 \left( \frac{I}{H_c} \right)^{1/3}, \tag{6}$$

   where $H_c$ is the heat of combustion of wildland fuels.

   (b) Size of firebrands: The convective activity inside the plume regulates the maximum size of the firebrand that can be lofted. The terminal velocities of the loftable firebrands can not exceed the vertical gas flow rate. As the vertical gas-flow increases with increasing fire rate, heavier firebrands can be uplifted into the plume. In literature, the maximum loftable radius for spherical firebrands is expressed as (Tarifa et al., 1965; Albini, 1979; Wang, 2011):

   $$r_{\text{max}} = \frac{3}{2} C_d \frac{\rho_a}{\rho_f} \frac{U_{\text{gas}}^2}{g}, \tag{7}$$

   where $\rho_a$ and $\rho_f$ represent the density of the ambient air and wild-land fuels respectively, $C_d$ is the drag coefficient and $g$ is the acceleration due to gravity.

   (c) Maximum loftable height: Wang (2011) and Woycheese et al. (1999) parametrize the maximum loftable height for spherical firebrands in terms of the radius of firebrand $r$, constrained by the maximum radius of the firebrands $r_{\text{max}}$:

   $$\mathcal{H} = 1.46 \left( \frac{\rho_f}{\rho_a C_d} \right) \frac{r_{\text{max}}^{5/2}}{r^{3/2}}. \tag{8}$$

2. Horizontal transport:

   (a) Maximum landing distance: Assuming the shape of the firebrands to be spherical, Tarifa et al. (1965) combines both experimental and theoretical approaches to characterise the maximum landing distances of the firebrands. Based on these results, Wang (2011) provides an approximation of the maximum travel distance for spherical firebrands from a vertical convective column in terms of the maximum loftable height $\mathcal{H}$, the meteorological mean wind $\mathcal{U} = |\mathbf{U_h}|$, where $\mathbf{U_h}$ is the horizontal wind vector field at fire-height, and the radius of the firebrands $r$:

   $$\mathcal{L} = \mathcal{H} \frac{\mathcal{U}}{U_{\text{gas}}} \left( \frac{r_{\text{max}}}{r} \right)^{1/2}. \tag{9}$$

3. Ignition probability: As described in the previous section, the probability that the fuel is ignited by burning embers is modelled using the function $\psi$ (2). Here we assume that the fuel conditions are homogeneous and the ignition probability depends only on an ignition delay $\tau$. No other local variables are taken into account.

4. Secondary fire-lines: The secondary emissions generated during the fire-spotting modelling are assumed as new sources of fire with a proper fire intensity. These new fires act as additional input along with the primary fire towards generation of other secondary fires, and it is assumed that these new sources are capable of generating firebrands of the same size as the primary source.

Small-scale processes, such as the mass loss of a firebrand due to combustion, affect the fire-spotting phenomenon by generating random fluctuations in the firebrand trajectory. This fluctuations are embodied by the use of a distribution for the landing distance.

Finally, the above large-scale subprocesses under lofting and transport mechanisms are linked through Eq. (5) to obtain the physical parametrisation of $\mu$ and $\sigma$ of the lognormal distribution. Using Eq. (9) and Eq. (8), we express the shape and location parameters as follows:

$$\sigma = \frac{1}{2z_p} \ln\left(\frac{\mathcal{U}^2}{rg}\right),$$
(10)

$$\mu = \mathcal{H}\left(\frac{3}{2}\frac{\rho_a}{\rho_f}C_d\right)^{1/2}.$$
(11)

We chose such parametrization of $\mu$ and $\sigma$ in order to de-lineate the governing parameters for lofting and transport mechanisms respectively. We hypothesise that the definition of $\mu$ covers the essential input parameters needed to describe the lofting mechanism of the firebrands inside the convective column. The relative density $\rho_a/\rho_f$ and atmospheric drag quantify the buoyant forces experienced by the firebrand; hence it is appropriate to include these quantities in definition of $\mu$. Substituting maximum loftable height from Eq. (8) in $\mu$ gives:

$$\mu = 3.52 \times 10^5 \left(\frac{\rho_a}{\rho_f}C_d\right)^2 \left(\frac{I}{H_c}\right)^{5/3} r^{-3/2}g^{-5/2}.$$
(12)

The radius of the firebrand $r$ and the fuel density are important ingredients to determine the height of the lofted firebrands. On the other hand, $\sigma$ is hypothesized to define the transport of firebrands under the effect of horizontal wind after ejection from the convective column. The definition of $\sigma$ includes a dimensionless ratio $\mathcal{F} = \mathcal{U}^2/(rg)$ which is analogous to the Froude number, which quantifies the balance between inertial and gravitational forces experienced by the firebrand. All firebrands with $r \leq \mathcal{U}^2/g$ can be transported by the horizontal wind.

In this model, the phenomenon of fire-spotting is assumed to occur together with the turbulent heat flux around the fire, and the turbulent diffusion coefficient $\mathscr{D}$ is utilised as a measure of the turbulent heat transfer generated by the fire. It is parametrised in terms of the Nusselt number $Nu$. Nusselt number defines the ratio between the convective and conductive heat transfer in fluids and is defined as $Nu = (\mathscr{D} + \chi)/\chi$ where $\chi = 2 \times 10^{-5}\,\mathrm{m^2s^{-1}}$ is the thermal diffusivity of air at ambient temperature. Experimentally, it is shown that Nusselt number is related to Rayleigh number as $Nu \simeq 0.1Ra^{1/3}$ (Niemela and Sreenivasan, 2006). Rayleigh number is defined as $Ra = \gamma \Delta T g h^3/(\nu\chi)$, where $\gamma = 3.4 \times 10^{-3}\,\mathrm{K^{-1}}$ is the thermal expansion coefficient, $h$ is the dimension of the convective cell, $\nu = 1.5 \times 10^{-5}\,\mathrm{m^2s^{-1}}$ is the kinematic viscosity and $\Delta T$ is the temperature gradient

between the top and bottom faces of the convective cell. Finally, we estimate the turbulent diffusion coefficient through the formula

$$\mathscr{D} \simeq 0.1 \left(\frac{\gamma g}{\nu}\right)^{1/3} \chi^{2/3} \Delta T^{1/3} h - \chi, \tag{13}$$

and assuming $\Delta T \simeq 1000\,\mathrm{K}$ and $h \simeq 100\,\mathrm{m}$ we have that $\mathscr{D} \sim 10^{-1}$. For all the simulations presented in this article, the value of the turbulent diffusion coefficient $\mathscr{D}$ is assumed to be $0.15\,\mathrm{m}^2\mathrm{s}^{-1}$.

The simple design of the physical parametrisation makes the model computationally less expensive and the requirement of defining only few vital parameters to execute any simulation also serves as an added advantage to the operational users. Static and dynamic input parameters of the model are reported in Table 1.

## 4   Numerical simulations

The detailed steps of the numerical procedure are as follows (Kaur et al., 2016):

1. Beginning with an initial fire-line, an operational code, i.e., `WRF-Sfire` in this Section and `LSFire+` in the next Section, is used to estimate the propagation of the front and to build up a new fire perimeter for the next time step. This output is modified to include the effects of turbulence and fire-spotting by a *post-processing numerical procedure*. This post-processing step is independent of the definition of ROS.

2. The fire perimeter obtained from the chosen operational code is used to construct the indicator function $I_\Omega(\boldsymbol{x}, t)$, i.e., the indicator function $I_\Omega(\boldsymbol{x}, t)$ has a value 1 inside the domain surrounded by the fire-line and 0 outside. The spatial information contained in $I_\Omega(\boldsymbol{x}, t)$ is necessary to modify the fire-line with respect to turbulence and fire-spotting and serves as an input to the *post-processing* step.

3. The effective indicator function $\phi_e(\boldsymbol{x}, t)$ (1) is generated over a Cartesian grid to facilitate the computation of the function $\psi(\boldsymbol{x}, t)$ (2) over the same grid.

4. The value of the effective indicator $\phi_e(\boldsymbol{x}, t)$ is computed through the numerical integration of the product of the indicator function $I_\Omega(\boldsymbol{x}, t)$ and the PDF of fluctuations according to (1). The effect of turbulence or fire-spotting is included by choosing the corresponding PDF (3).

5. The function $\psi(\boldsymbol{x}, t)$ is updated for each grid point by integration in time with the current value of $\phi_e(\boldsymbol{x}, t)$.

6. All points which satisfy the condition $\psi(\boldsymbol{x}, t) \geq 1$ are labelled as new ignition spots. The *post-processing* procedure is completed at this step.

7. At the next time step, the new fire perimeter evolves according to the chosen operational code, and the updated perimeter is again subjected to the *post-processing* proceedure to enrich the fire front with the random fluctuations pertaining to turbulence and fire-spotting. The sequence is repeated till the final "event time" step or till the fire reaches the boundaries of the domain.

## 4.1 Implementation of `RandomFront 2.3` in `WRF-Sfire`

In order to prove the viability of the proposed formulation within an operational code, we have implemented `RandomFront 2.3` in the framework of the `WRF-Sfire` simulator (Coen et al. (2012); Mandel et al. (2014)). `WRF-Sfire` is a coupled fire-atmosphere model, which operates in the computational environment of a well known public domain numerical weather prediction model: `WRF v3.4` (Weather Research and Forecasting; (Skamarock et al., 2008); http://wrf-model.org/users/users.php)

The fire module embedded into `WRF` simulates a surface fire and takes into account a two-way coupling with the atmospheric model. The near-surface winds from the atmospheric model are interpolated on the fire grid and are used along with fuel properties and local terrain gradients to compute both the ROS and the outward front-direction, that are further used as an input to the front propagation routines through a LSM scheme. Fuel consumption is responsible for the release of sensible and latent heat into the lowest layers of the atmosphere and this has a role in the computation of the boundary-layer meteorology. Recently, the model has been equipped with a fuel-moisture sub-model and a chemistry sub-model (`WRF Chem`), which contribute towards reproducing and investigating the effects of the fire-atmosphere coupling.

Coen et al. (2012) points out that fires generally start from a horizontal extent much smaller than the size of the fire mesh-cell. The same may be argued for the secondary ignitions related to fire-spotting phenomenon. In this respect, Coen et al. (2012) propose and explain in detail an algorithm for a punctual or line ignition that actually runs on `WRF-Sfire`. The purpose of this algorithm is two-fold: $i$) it guarantees from a physical point of view that the ignition starts at sub-grid scale without generating unrealistically large initial heat flux and an accelerated ignition; $ii$) this procedure is numerically robust because it is fully integrated into the representation in terms of a *signed distance function* of the LSM (Sethian and Smereka, 2003).

In the proposed formulation, a punctual ignition occurs whenever the condition $\psi(\mathbf{x}, t) \geq 1$ holds true. This procedure is not computationally viable so we set a threshold distance $R_{\mathrm{th}} = 200\,\mathrm{m}$ for separating each pair of punctual ignitions. In particular, let $\mathcal{P}$ be the set of point-wise fire-spotting ignitions, the actual algorithm performed at each time-step within `WRF-Sfire` model is reported in Algorithm 1.

---

**Algorithm 1** Algorithm for Point-Wise ignition due to Fire-Spotting

---

1: **for** $\mathbf{x}_i \in$ Grid **do**
2:    **if** $\psi(\mathbf{x}_i, t) > 1 \wedge \mathbf{x}_i \notin \Omega(t)$ **then**
3:       **if** $\mathrm{dist}(\mathcal{P}, \mathbf{x}_i) > R_{\mathrm{th}}$ **then**
4:          New fire-spotting ignition in $\mathbf{x}_i$ **and**
5:          $\mathcal{P} \leftarrow \mathcal{P} \cup \{\mathbf{x}_i\}$
6:       **end if**
7:    **end if**
8: **end for**

---

### 4.2 Discussion of the test case with `WRF-Sfire`

In this paper we consider a slight modification of the `hill` test case (https://github.com/openwfm/wrf-fire/blob/master/wrfv2_fire/test/em_fire/hill/namelist.input.hill). In order to simplify the underlying dynamics, but keeping the fire-atmosphere coupling, the hill is suppressed (`fire_mountain_type` $= 0$) and we have a square-grid simulation over a flat domain with side

2.5 km. The horizontal atmospheric grid-spacing at terrain-height is 60 m, while the fire spread grid-spacing is 15 m. The simulation starts at the instant $t = 0$ min and ends at $t = 20$ min. The fire-line is initially located along the segment joining the points $(1900\,\mathrm{m}, 1500\,\mathrm{m})$ and $(1900\,\mathrm{m}, 1800\,\mathrm{m})$, and the initial wind field at the fire height is $(U(\mathbf{x}; t = 0), V(\mathbf{x}; t = 0)) = (-6.4\,\mathrm{ms}^{-1}, -3.6\,\mathrm{ms}^{-1})$ in all the points of the simulation domain.

     The fuel has been set equal to fire *Type 9*, i.e., *FM 9 Hardwood litter* according to Anderson classification (Anderson, 1982).

This fuel type may represent a terrain covered by *Pinus ponderosa* trees. The radius of the spherical embers has been set equal to $r = 12.5$ mm following a size considered by Manzello et al. (2006) with the same vegetation.

     For what concerns the proposed formulation, the fire-line intensity $I$ and the wind field are computed by means of the `WRF-Sfire` model, and this allows for a varying field of both parameters $\sigma$ and $\mu$ according to formulae (10) and (12), respectively. In particular, following latest advancements of `Sfire` environment in Mandel et al. (2014), spatial representation

of the potential-fire characteristics are available from which a field extension of the fire-line intensity $I$ is available in order to have a space and time varying field of $\mu$ for the fire-spotting routines. Parameters $\mathscr{D}$ and $\tau$ are set as $\mathscr{D} = 0.15\,\mathrm{m}^2\mathrm{s}^{-1}$ and $\tau = 8$ s, without using estimations by `WRF-Sfire`.

     Figures 1-3 display the simulation results. In each figure the firefront is reported by a dashed line at the following instants $t = 6$ min, $t = 10$ min and $t = 20$ min. The selected instants allow for observing the propagation of the main fire alone, the

generation of a secondary fire and the multi-generation of secondary fires. In particular, in Fig. 1 the evolution of the fire-line is shown in relation to the three components of the wind field; Fig. 2 shows the relationship with parameter $\mu$ and the fire intensity field; Fig. 3 shows the relationship with parameter $\sigma$ and the squared norm of the horizontal wind.

     Overall, we observe that the fire-line propagation is "pulled" in the direction of the maximum value of the squared norm of the horizontal wind (see right column in Fig. 3), and this direction is induced by the fire itself as a feedback on the weather as it

is shown by the patterns of the atmospheric observables when secondary fires are generated. The geometrical profile of the fire perimeter always plays an important role in determining the fire-spotting behaviour. The asymmetry in the fire perimeter at 20 min along the prominent direction of propagation, causes the first secondary fire to appear in the top-left part of the domain. With time, as the fire-activity increases, the differences between the maximum value of the squared norm of the horizontal wind and its surroundings increase and the fire-line becomes symmetric with respect to the main direction of propagation. This

has a direct influence of the fire-spotting action, and the new secondary fires appear increasingly aligned towards the main direction of propagation.

     The secondary fires are equally important as the primary fire in influencing the weather around the fire. The plots clearly show the influence of fire-atmosphere coupling, and a feedback dynamics from secondary fires to primary fire can be also observed. The secondary fires affect the wind (see Fig. 1) and also the parameter $\sigma$ (Fig. 3), which implies a refinement in fire-spotting

characteristics for further ignitions. A point worth noting is that the first secondary fire occurs at a distance of almost $1500\,\mathrm{m}$ from the main fire. This observation supports the viability of the proposed formulation to simulate fire-spotting mechanism in studies of large-scale fires. However, if the implementation in `WRF-Sfire` allows for a comprehensive picture including the physical features of a multi-scale and multi-physics process, the complexity of the model, the number of parameters and the numerical cost increases.

In the next section, we perform a response analysis of our parametrisation, by using the simple finite difference code `LSFire+`, that allows for extensive simulations in terms of spatial domain and simulation time. A sensitivity analysis on the inputs and an uncertainty quantification on the outputs of `RandomFront` implemented in `LSFire+` will be considered in a separate paper.

## 5  Response analysis

### 5.1  Implementation on `RandomFront 2.3` into `LSFire+`

A few idealized simulations are carried out to highlight the potential applicability of the formulation. For all the simulations, a flat domain with a homogeneous coverage of *Pinus ponderosa* ecosystem is selected, as seen in the `WRF-Sfire` implementation. The simulations are run using a basic set-up of wildfire model `LSFire+` which involves a moving interface method based on the LSM (Pagnini and Massidda, 2012a, b; Pagnini and Mentrelli, 2014). The Byram formula (Byram, 1959; Alexander, 1982) is used to estimate the ROS of the fire-line:

$$\mathcal{V}_{ros}(\boldsymbol{x},t) = \frac{I(1+f_w)}{H\,\alpha\,\omega_0},\tag{14}$$

where $I$ is the fire intensity, $H = 22000\,\mathrm{kJ\,kg^{-1}}$ is the fuel low heat of combustion, $\omega_0 = 2.243\,\mathrm{kg\,m^{-2}}$ is the oven-dry mass of the fuel and the functional dependence on the wind is included through the factor $f_w$. The user has flexibility to introduce a different ecosystem in the simulations by modifying the the parameters $H$ and $\omega_0$. The parameter $\alpha$ is chosen to guarantee that the maximum ROS is always equal to the ROS prescribed by the Byram formulation.

The response of the formulation to depict the different firebrand landing distributions is highlighted through two sets of test cases. In the first test case, the wind conditions and the size of the firebrands are assumed to be constant as the fire intensity changes. In the second test case, the fire intensity is assumed to be constant and the simulations for different wind conditions are carried out. The second test case is also repeated for different firebrand radii.

For speeding-up all simulations presented in this Section, the domain has been scaled by a factor of 4 to reduce the computation time. In the scaled mode each grid cell represents $4 \times 4$ time the area of each grid cell of the original domain. This scaling also affects the ROS and the turbulent diffusion coefficient and their value is reduced by a factor of 4. Fire intensity and the wind speed remain unaffected by the re-scaling. It is mentioned that such scaling has no effect on the outputs of the simulations but helps in reducing the computation time.

It is remarked that, in the simulations presented in this paper, the firebrands are considered to be a sphere of constant radius for each simulation; but in real situations all shapes and sizes of the firebrand are produced from the fuel. It is also emphasized

that the selection of the domain and other parameters do not correspond to any real fire but an effort is made to chose the values of different parameters to lie in the valid range.

## 5.2 Discussion for `LSFire+`

The mathematical formulation of the random effects presented in this article considers the effects of turbulence and fire-spotting together. The main highlight of this formulation is its ability to incorporate the generation of secondary fires. The top panel of Fig 4 shows the evolution of fire-perimeter under the effect of turbulence and fire-spotting. The figure on the left shows the effect of fire-spotting in the presence of a barrier. This barrier is a fuel free zone with zero probability of ignition. The fire break zone stops the spreading of the fire, but at 50 minutes, a new secondary fire appears beyond the barrier. This new fire-line is completely detached from the main parent line though it originates from the fire-spotting effects of the main fire-line. The parent fire shows a negligible growth in the head and cross wind direction, but the secondary fires grow up quickly under the effect of wind. The wavy pattern of the fire-perimeter results from the merging of multiple secondary fires in an idealized set-up of constant conditions. The figure on the right shows the scenario where secondary fires appear without the presence of a barrier. The effect of fire-brands first appears around 130 minutes and henceforth, the new secondary fire behaves as a separate fire and evolves accordingly. In `LSFire+` the effects of firespotting occur in conjunction with turbulence and both processes contribute towards the fire propagation. It is difficult to separate the effects of both processes individually, but a comparison of the increase in burned area due to turbulence and turbulence + firespotting is presented in bottom panel of Fig. 4. The total number of burned points is plotted at different times for two simulations: only turbulence, and turbulence + fire-spotting. All the simulation parameters remain the same in both the simulations. As the fire starts evolving, the line plots for both the simulations overlap signifying that fire-spotting has no visible contribution, but after 50 minutes, the fire-spotting effect picks up and the burned area increases rapidly. At 140 minutes increase in the burned area under the combined effect of the two random processes is almost three times the effect of turbulence alone.

For the response analysis separate set of simulations are carried and the response is evaluated through a parameter $\beta_e$, which describes the effective increase in the burned area:

$$\beta_e = (\mathbf{x}_{\mathrm{random}} - \mathbf{x}_{\mathrm{no-random}})/\mathbf{x}_{\mathrm{no-random}}. \tag{15}$$

$\beta_e$ is simply the increase in the number of burned grid points with respect to the simulation when no random effects are considered.

The simulated domain for the response analysis is chosen as a rectangle of dimensions $[0\mathrm{m}, 6000\mathrm{m}] \times [0\mathrm{m}, 6000\mathrm{m}]$. The simulations start at time $t = 0\,\mathrm{min}$ and end at time $t = 140\,\mathrm{min}$. The grid spacing is $\Delta x = \Delta y = 20\mathrm{m}$. At time $t = 0\,\mathrm{min}$ the initial fire-line is a circle of radius $180\,\mathrm{m}$ centered at $\mathbf{x}_c = (720\mathrm{m}, 3000\mathrm{m})$. The horizontal wind has been assumed in this simulation set-up as a constant field parallel to the vector $\mathbf{j} = (1, 0)$ and with modulus $|\mathbf{U_h}| = |(U, V)|$.

## Response analysis to fire intensity

An increase in the fire intensity causes an increase in the burned area (see Eqn (14) for the definition of the ROS); at the same time, the fire-spotting behaviour is also affected by any change in $I$. The parameter $\beta_e$ allows us to identify the contribution of fire-spotting towards the fire-propagation. The top panel of Fig. 5 shows the change in the burned area under the combined effect of turbulence and fire-spotting with increase in the fire-intensity. The two line plots correspond to a constant wind speed ($10\,\mathrm{ms}^{-1}$) but two firebrand radii, i.e., $0.015\,\mathrm{m}$ and $0.030\mathrm{m}$. According the physical parametrisation of the lognormal shape parameters $\mu$ and $\sigma$, for these set of simulations (increase in fire intensity $I$), the parameter $\mu$ varies while parameter $\sigma$ remains constant. For the both set of radii, an increase in the fire intensity shows an sharp increase in the burnt area for low fire intensities. A zoom-in of this sharp rise is also shown in the top right figure. For smaller firebrand radius, the fire-spotting effect shows a slight saturation between 15-25 $\mathrm{MWm}^{-1}$, but with further increase in the fire intensity, the contribution of fire-spotting remains positive till $60\mathrm{MWm}^{-1}$ and then it saturates again. Any further increase in the fire-intensity causes a decreasing importance of the fire-spotting. For the larger firebrand radius, a similar behavior is observed for fire intensities less than $15\mathrm{MWm}^{-1}$, but with further increase in $I$, the contribution from fire-spotting takes a dip before it starts to increase again. A zoom-in of the response analysis is important as literature shows that for fire intensities around $8\mathrm{MWm}^{-1}$ for vegetation type *Pinus ponderosa* are classified as high ""severity class" (Chatto and Tolhurst, 2004). For fire intensities ranging upto 10 $\mathrm{MWm}^{-1}$ we can observe a rapid increase in the fire-spotting behaviour upto $4\mathrm{MWm}^{-1}$. Any further increase in the fire intensity has a positive effect on the ROS and on the propagation of the main fire, such that less effect on the fire-line due to fire-spotting are observed. It is also interesting to note that for weak fires (less than $1\mathrm{MWm}^{-1}$), the fire-spotting mechanism is independent of the firebrand radius.

In order to explain these observations, we plot the lognormal distribution for selected values of $I$ (bottom panel, Figure 5). These lognormal distribution plots show general trend in the distribution with varying $\mu$ but constant $\sigma$. With an increase in $I$ (or $\mu$), the maximum probability increases but the distribution becomes increasingly skewed. For this particular set-up of parameters $\mu$ and $\sigma$ the skewness is more pronounced for fire intensities greater than $20\mathrm{MWm}^{-1}$ (bottom right). For lower values of $I$ (less than $20\mathrm{MWm}^{-1}$), the lognormal distribution tapers off slowly and the probability of a "long-range" ignition increases. This explains large initial increase in the contribution of the fire-spotting behaviour for low range of fire-intensities. As the as the magnitude of the corresponding peak value is also decreasing with increasing $\mu$ or $I$, this "long-range" ignition probability can have a positive influence only till a certain threshold. This threshold is the range where parameter $\beta_e$ takes a dip or shows a saturation. Beyond this point, the effective contribution of "long-range" probability diminishes, and the contribution of the"short-range" ignitions becomes increasingly important. The gradual increase in the effective burned area for both firebrand sizes (fire-intensities greater than 30 $\mathrm{MWm}^{-1}$) can be attributed to this reason. For large values of $\mu$ the lognormal distributions tend to be similar though they retain their behaviour of becoming increasingly skewed (figure bottom right). Ideally, with increasing skewness, the "short-range" probability will lie much closer to the main fire-line and the effective contribution of fire-spotting should decrease. But since we observe such behaviour only for the smaller firebrand, such behaviour may exist for the large firebrand only outside the current range of simulations.

The effect of the fire intensity on the fire-spotting behaviour can also be explained by the physical parametrisation provided in this article. According to the physical parametrisation proposed here, an increase in the fire intensity increases the maximum loftable height, hence the firebrands are ejected from elevated heights. Higher release height contributes towards an increase in the firebrand activity at longer distances and the initial increase in the fire perimeter follows this observation. But at the same instance, the increase in the firebrand ejection height over constant wind conditions causes the firebrands to travel longer in the atmosphere before hitting the ground. The growing travel time for a firebrand promotes its combustion and the firebrand reaches the ground with a lower temperature (less "long-range" ignition probability) than its counterpart ejected at lower heights. Lower temperature of the firebrands leads to an inadequate heat exchange with the unburned fuel for a successful ignition and hence after reaching an area of maximum activity, the effective contribution of the "long-range" firebrands under same atmospheric conditions diminishes with increasing fire activity. This explains the initial dip/saturation in the fire-brand activity. At the same instance, the "short-range" firebrands have larger energy and becomes the dominant cause of the fire-spread. This range can be considered as the transition time when the "long-range" firebrands become less important but the "short-range" activity picks up. For the heavier fire-brands a similar behaviour is expected, though the maximum loftable height under identical wind and fire conditions is lower. Lower loftable height decreases the maximum landing distance and the magnitude of the burned area is less, which is also evident from the lower magnitude of $\beta_e$ in the results.

**Response analysis to wind speed**

Bottom panel of Fig. 6 highlights the simulation results with an increasing value of the wind velocity over constant fire intensity ($50\,\mathrm{MWm}^{-1}$). The results for two different radii ($0.015\,\mathrm{m}$ and $0.030\,\mathrm{m}$) are presented. The fire-spotting mechanism over varying wind speeds shows similar behaviour for different size of the firebrands. For both radii, the effective burnt area increases with the increasing wind speed but after a certain threshold, an increase in the wind speed leads to a decline in the effective burnt area. The line plot for $r = 0.015\,\mathrm{m}$ shows that after a value around $10\,\mathrm{ms}^{-1}$, the contribution of the firebrands decreases. Similarly, in the blue line-plot for radius $r = 0.030\,\mathrm{m}$, the effective increase in the burned area follows an identical pattern but the total increase in the burned area is lesser in magnitude and shows a saturation around the maxima before it starts to fall (around $22\mathrm{ms}^{-1}$). For bigger firebrands the onset of the maximum occurs at higher wind speeds and it sustains longer.

The lognormal distributions for selected values of wind speed (but fixed $r$ and $I$) are plotted in the bottom panel of Figure 6. These two plots show two different aspects of the response behaviour of the lognormal distribution when parameter $\sigma$ is varying but parameter $\mu$ is constant. Firstly, from the bottom left figure it is evident that with increase in the wind speed (increasing $\sigma$, constant $\mu$) the lognormal distribution shifts towards the left but the tails taper off slowly, making the distribution wider around the maximum. The increase in the width of the lognormal distribution leads to a larger area of "long-range" and "short-range" probability and hence explains the initial rise of the burned area in the top panel. At the same time, the increasing wind speed also causes a decrease in the magnitude of the probability, hence beyond a certain threshold, the overall contribution from fire-spotting starts to fall. The saturation in the fire-spotting behaviour can be explained by the second aspect of the lognormal response. The figure on the bottom right shows that after a certain threshold of parameter $\sigma$ or wind speed, the lognormal distributions become increasingly similar. This threshold depends upon the value of parameter $\mu$, and for smaller values of

$\mu$ (smaller $r$ or $I$), we have an early onset of this threshold. As the lognormal distributions tend to have similar probability distributions with increasing wind speed (figure bottom right), the contribution from fire-spotting also becomes similar and it explains the saturation in the fire-spotting behaviour of bigger firebrand radius.

This response of the model over different wind velocities can also be explained through physical parametrisation of $\sigma$. In terms of the physical quantities used in the parametrisation, it can be argued that strong winds can carry away the firebrands at longer distances from the main source and result in a larger fire perimeter (increasing "long-range probability"). Historically it has been reported that strong winds coupled with extremely dry conditions formed the perfect recipe for long range fire-spotting. Strong wind speeds can loft the smaller firebrands to longer distances but with an increasing wind speed the combustion process quickens and the firebrands reach the ground with less temperature. This fact explains the reduced effect of fire-spotting on the burned area over high wind conditions. On the other hand, a larger firebrand size can sustain longer in the atmosphere hence their "long-range" probability is relatively higher than for smaller firebrands. This explains the onset of maximum burned area at $15 \text{ms}^{-1}$ instead of $12 \text{ ms}^{-1}$ for 0.015m radius. The heavier mass of bigger firebrands restricts their flight to shorter distances in comparison to the lighter firebrands and hence a lower magnitude of the burned area is also observed. The longer saturation in the fire-brand activity for the larger firebrand is because of its ability to stay longer in air without burning out.

## 6 Conclusions

A mathematical formulation complete with its physical parametrisation `RandomFront 2.3` to reproduce/mimic the fire-spotting behavior is presented in this article. In particular, we provide a versatile probabilistic model for fire-spotting that is suitable for implementation as a post-processing scheme at each time step in any of the existing operational large-scale wildfire propagation models, without calling for any major changes in the original framework. This simple physical parametrisation is also an added advantage for real-time application. In this respect, `RandomFront 2.3` is implemented in the coupled fire-atmosphere model `WRF-Sfire` and a simple test case discussed. Moreover, the proposed formulation and parametrisation can be extended to include further variables such as moisture, spatial distribution of combustible, orography and also atmospheric variables such as wind and pressure that are available by running the simulation with `WRF-Sfire`. This allows for shading light on the interaction between the concurrent factors in wildfires.

Furthermore, simulations with a simple propagation model based on the Level Set Method, namely `LSFire+`, are performed to highlight the different responses of the model towards varying fire intensities and wind conditions, and constant climatic conditions are assumed in the entire set-up. Results with different firebrand radii are also shown.

In both implementations, i.e., `WRF-Sfire` and `LSFire+`, the simulations are simplified to highlight the physical applicability of the model.

The parametrisation `RandomFront 2.3` provides a simple yet versatile addition to operational fire spread models reproducing the different physical aspects of the firebrand landing behavior and simulating the occurrence of secondary fires as a result. The new secondary fires are also capable of modulating the weather around the primary fire and clear interactions

between the two are observed. The wind conditions and fire-perimeter play an important role to determine the occurrence of the secondary fires.

The results highlight that the parametrisation is successful in reproducing the different physical aspects of the firebrand landing behavior. In this model, the complexities related to the shape and density of the firebrands are not considered and for brevity they are assumed to be spherical with the diameter of the order of the "collapse diameter". The model also does not include an explicit computation of the time taken to reach the charred oxidation state, but an heating-before-burning mechanism is introduced in the mathematical formulation to serve a similar purpose. The inferences made from the simulations clearly fit within the physical aspects of the fire-spotting process. The increase in the wind speed causes an initial flare up in the fire perimeter, but in really high wind conditions, the wind enhances the propagation of the main fire and this reduces the effective contribution of the fire-spotting. Similarly, with increasing fire intensities, the fire intensity enhances the ROS of the main fire such that new ignitions of the unburned fuel ahead of the main fire by fire-spotting are reduced.

Though many other studies focus on the long range landing distributions of the firebrands, most of them include rigorous computational aspects like LES which limit their applicability to the operational models for wildfire propagation. The simple yet powerful probabilistic formulation presented in this paper obeys the physical aspects of the fire-spotting process and provides scope for its applicability to operational fire spread models.

*Code availability.* The code `LSFire+` is developed in C and Fortran where the model `RandomFront 2.3` acts as a post-processing routine at each time step in a LSM code for the front propagation implemented by the means of `LSMLIB` (Chu and Prodanović, 2009) and the ROS is computed by using the library `FireLib` (Bevins, 1996). The numerical library `LSMLIB` is written in Fortran2008/OpenMP and propagates the fire-line through standard algorithms for the LSM, including also Fast Marching Method algorithms. Furthermore, the routine `RandomFront 2.3` has been also implemented in the latest released version of `WRF-Sfire` (https://github.com/openwfm/wrf-fire/) by introducing new ad-hoc routines. Both implementations of `RandomFront 2.3` in `LSFire+` and `WRF-Sfire` are freely available at the official git repository of BCAM, Bilbao, https://gitlab.bcamath.org/atrucchia/randomfront-wrfsfire-lsfire.

Simulations with `WRF-Sfire` have been perfomed on a Intel(R) Core(TM) i5-4310M 2.70GHz CPU laptop with 8 GB of RAM. Each simulation that spanned 20 physical minutes took about 100 minutes of computational time.

Simulations with `LSFire+` are perfomed over the cluster HYPATIA of BCAM, Bilbao, using OpenMP shared memory parallelism, running over 24 cores inside of a Intel(R) Xeon(R) CPU E5-2680 v3 2.50GHz node with 128GB RAM. The computational time for each simulation, that spanned 140 minutes of physical time, was about 45 minutes.

The $80\%$ of the computational cost in both cases, i.e., `WRF-Sfire` and `LSFire+`, is due to the post-processing routine `RandomFront 2.3`. This computational time can be reduced in the future through a further code optimisation.

*Author contributions.* The research was planned and coordinated by GP in collaboration with IK. GP and IK formulated the physical parametrisation. IK performed the simulations of the test cases with `LSFire+` and gave the main contribution to writing the text. AT

implemented the routines of `RandomFront 2.3` into `WRF-Sfire` and performed the corresponding simulations. AB contributed to run the simulations with `LSFire+` and VE contributed to run the simulations with `WRF-Sfire`.

*Competing interests.* The authors declare that they have no conflict of interest.

*Acknowledgements.* This research is supported by the Basque Government through the BERC 2014–2017 and BERC 2018–2021 programs
and by the Spanish Ministry of Economy and Competitiveness MINECO through BCAM Severo Ochoa accreditation SEV-2013-0323 and
through projects MTM2013-40824-P "ASGAL" and MTM2016-76016-R "MIP", and by the PhD grant "La Caixa 2014". The research started
and mainly developed at BCAM, Bilbao, during the postdoc fellowship of IK and an internship period of AB.

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

**Table 1.** List of symbols.

| Model quantities | Units |
|---|---|
| $\phi_e$, effective indicator | - |
| $\psi$, ignition function | - |
| $\boldsymbol{x} = (x, y)$, horizontal space variable | m |
| $t$, time | s |
| $f$, probability density function of the random processes | $\mathrm{m}^{-2}$ |
| $G(\boldsymbol{x}; t))$, isotropic bi-variate Gaussian probability density | $\mathrm{m}^{-2}$ |
| $q(l)$, lognormal distribution of firebrand landing | $\mathrm{m}^{-1}$ |
| **Parameters** | |
| **Static Parameters** | **Value** |
| $\mu$ , parameter of $q(l)$ | – |
| $\sigma$, parameter of $q(l)$ | – |
| $\mathscr{D}$ turbulent diffusion coefficient of $G$ | $0.15 \mathrm{m}^2\mathrm{s}^{-1}$ |
| $\rho_a$, density of air | $1.2\ \mathrm{kgm}^{-3}$ |
| $\rho_f$, Density of wildland fuel (*Pinus Ponderosa*) | $542\ \mathrm{kgm}^{-3}$ |
| $C_d$, drag coefficient | 0.45 |
| $z_p$, p-th percentile | 0.45 |
| $g$, acceleration due to gravity | $9.8\ \mathrm{ms}^{-1}$ |
| $H_c$, heat of combustion of wildland fuels | $18620\ \mathrm{kJkg}^{-1}$ |
| $\omega_0$, oven-dry mass of fuel | $2.243\ \mathrm{kgm}^{-2}$ |
| $H$, fuel low heat of combustion | $22000\ \mathrm{kJkg}^{-1}$ |
| **Dynamic Parameters** | **Units** |
| $\mathbf{U} = (U, V, W)$, wind vector at fire-height | $\mathrm{ms}^{-1}$ |
| $\mathbf{U_h} = (U, V)$, horizontal wind vector field at fire-height | $\mathrm{ms}^{-1}$ |
| $\tau$, ignition delay of firebrands | s |
| $I$, fire-line intensity | $\mathrm{MWm}^{-1}$ |
| $U_{\mathrm{gas}}$, Vertical gas flow | $\mathrm{ms}^{-1}$ |
| $r$, radius of spherical firebrand | m |
| $r_{\mathrm{max}}$, maximum loftable radius for spherical firebrand | m |
| $\mathcal{H}$, maximum loftable height for spherical firebrands | m |
| $\mathcal{V}_{ros}$, rate of spread | $\mathrm{ms}^{-1}$ |

| Quantity | Unit of measurement | First Test Case | Second Test Case |
|---|---|---|---|
| $\mathscr{D}$ | $\mathrm{m^2 s^{-1}}$ | 0.15 | 0.15 |
| $U$ | $\mathrm{ms^{-1}}$ | 10 | $2 \div 26$ |
| $I$ | $\mathrm{MWm^{-1}}$ | $5 \div 100$ | 50 |
| $r$ | m | 0.015 | $0.015 \div 0.03$ |
| $\tau$ | s | 1 | 1 |

**Table 2.** Values of the main parameters for numerical simulations performed with `LSFire+`.

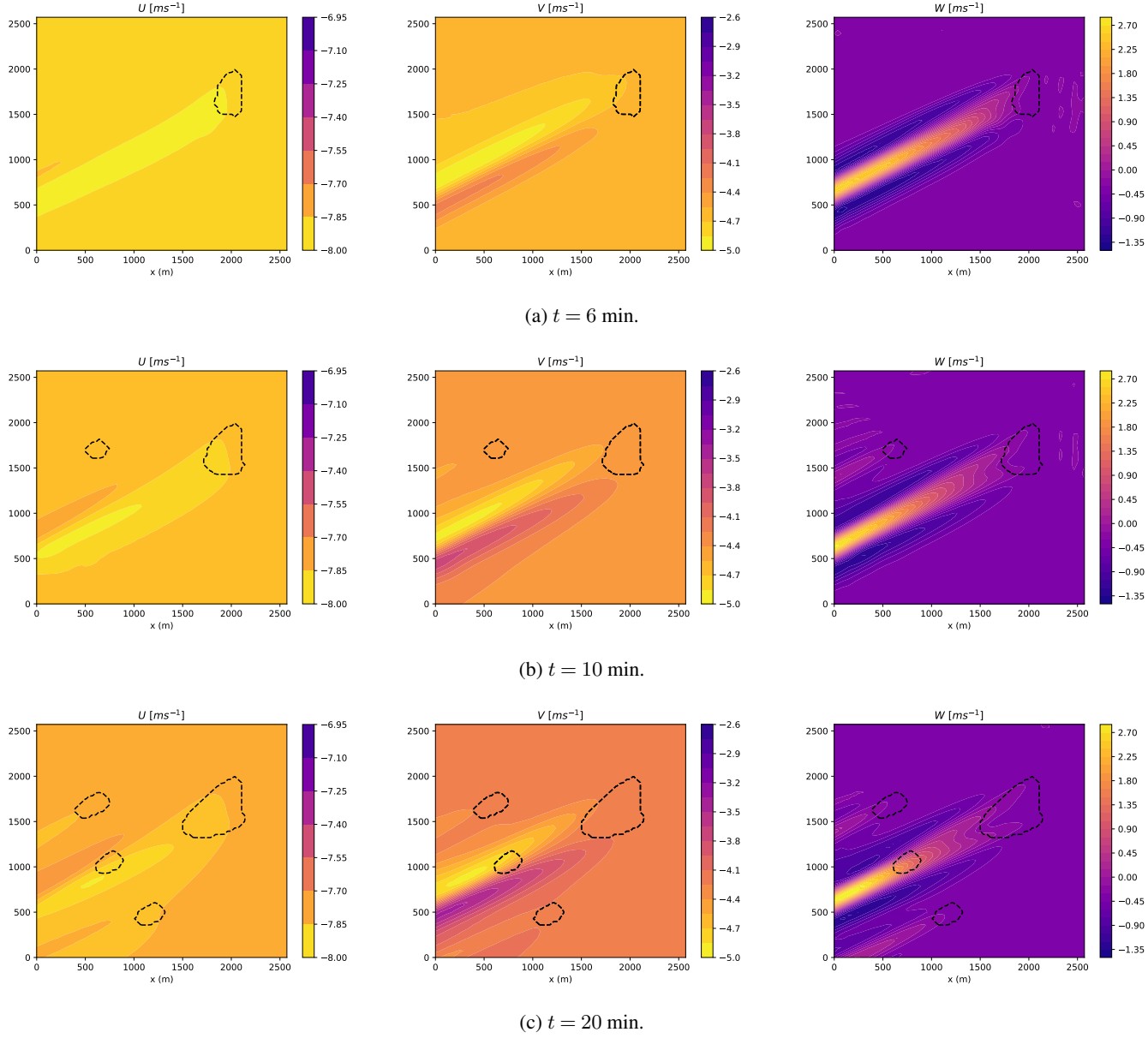

**Figure 1.** Wind vector components $(U, V, W)$ performed with `WRF-Sfire` at times $t = 6, 10, 20$ min. Firefront is reported by a dashed line.

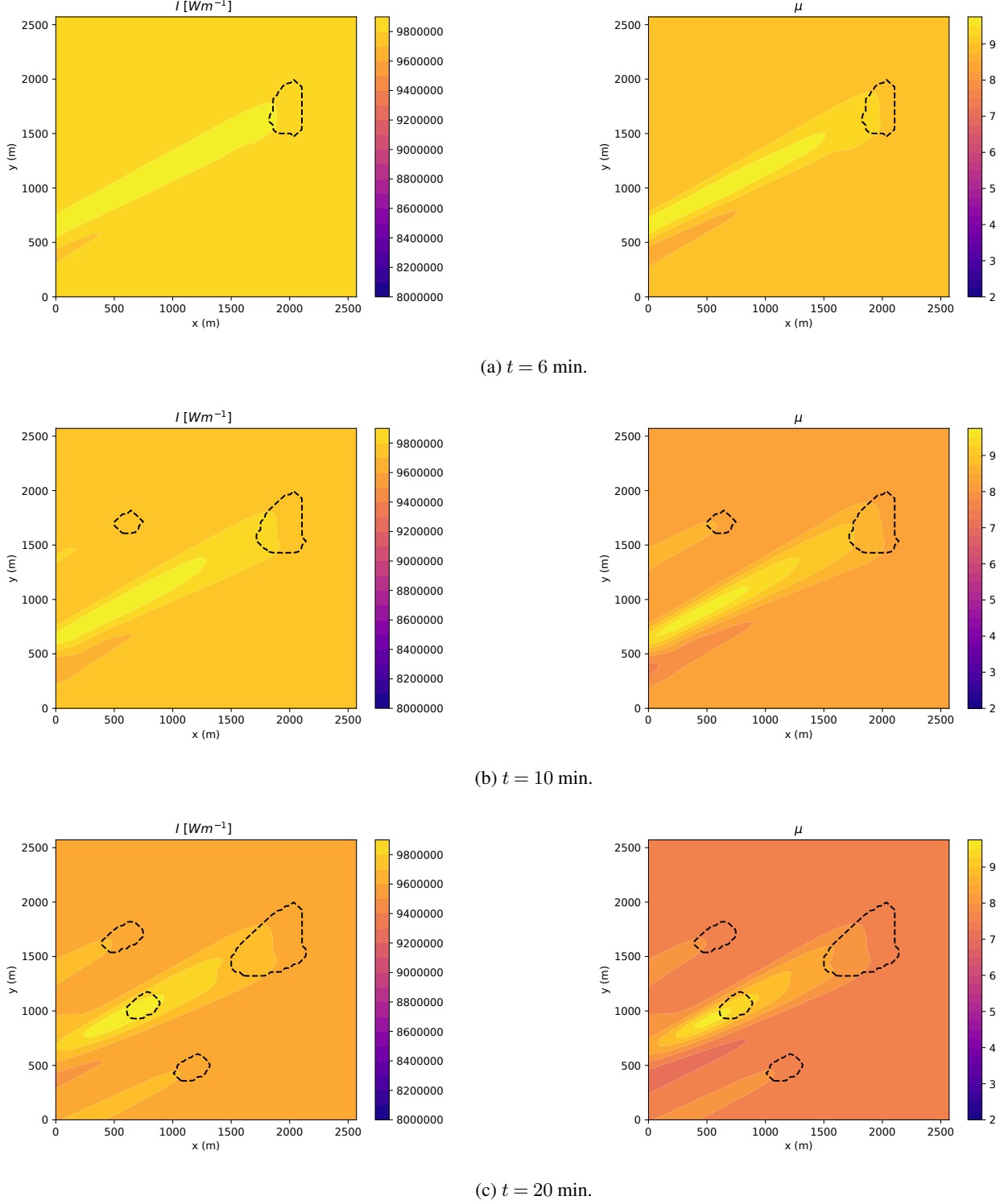

(a) $t = 6$ min.

(b) $t = 10$ min.

(c) $t = 20$ min.

**Figure 2.** Fire intensity $I$ and PDF shape parameter $\mu$ performed with `WRF-Sfire` at times $t = 6, 10, 20$ min. Firefront is reported by a dashed line.

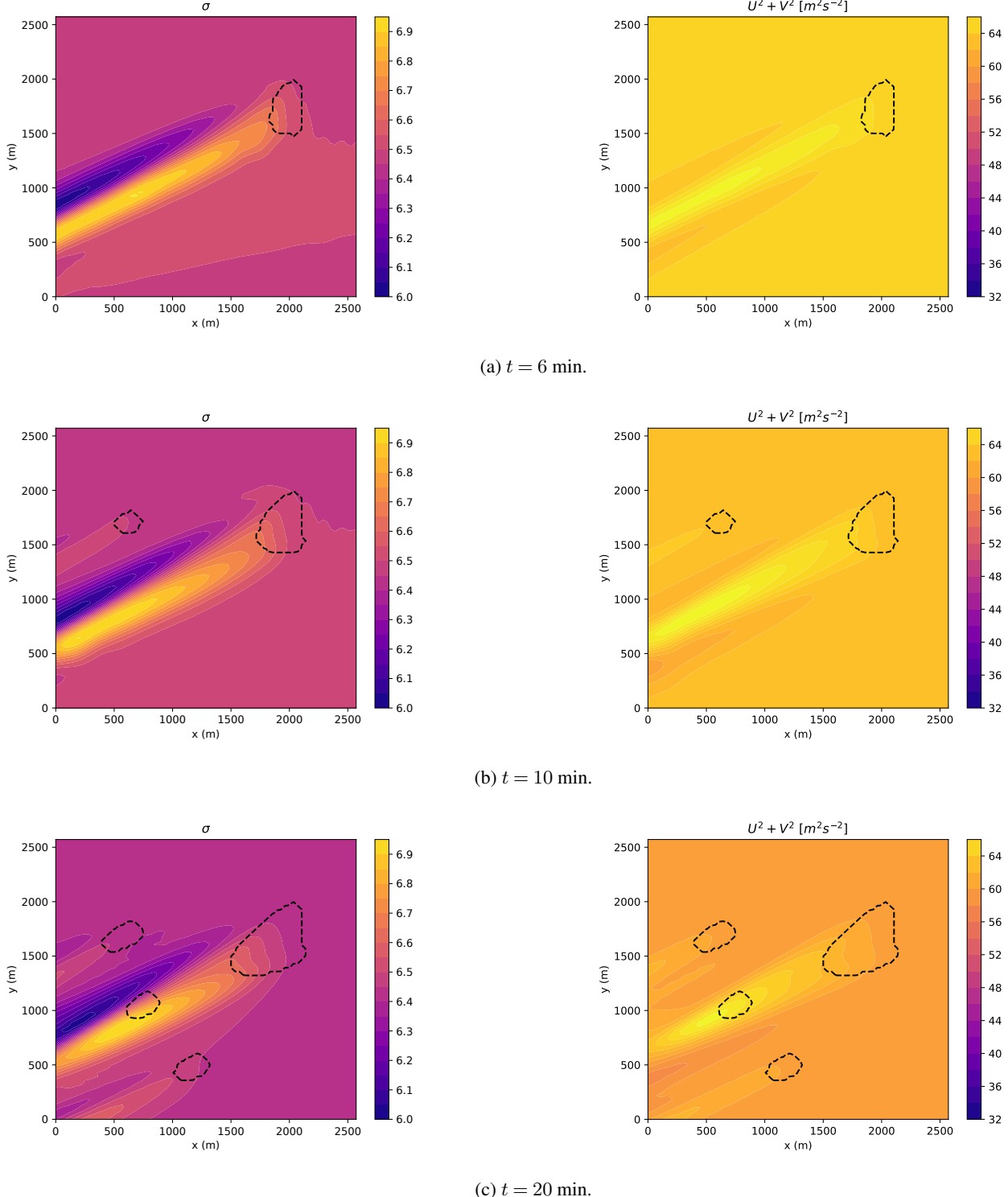

(a) $t = 6$ min.

(b) $t = 10$ min.

(c) $t = 20$ min.

**Figure 3.** PDF shape parameter $\sigma$ and horizontal wind squared magnitude ($|\mathbf{U_h}|^2$) performed with `WRF-Sfire` at times $t = 6, 10, 20$ min. Firefront is reported by a dashed line.

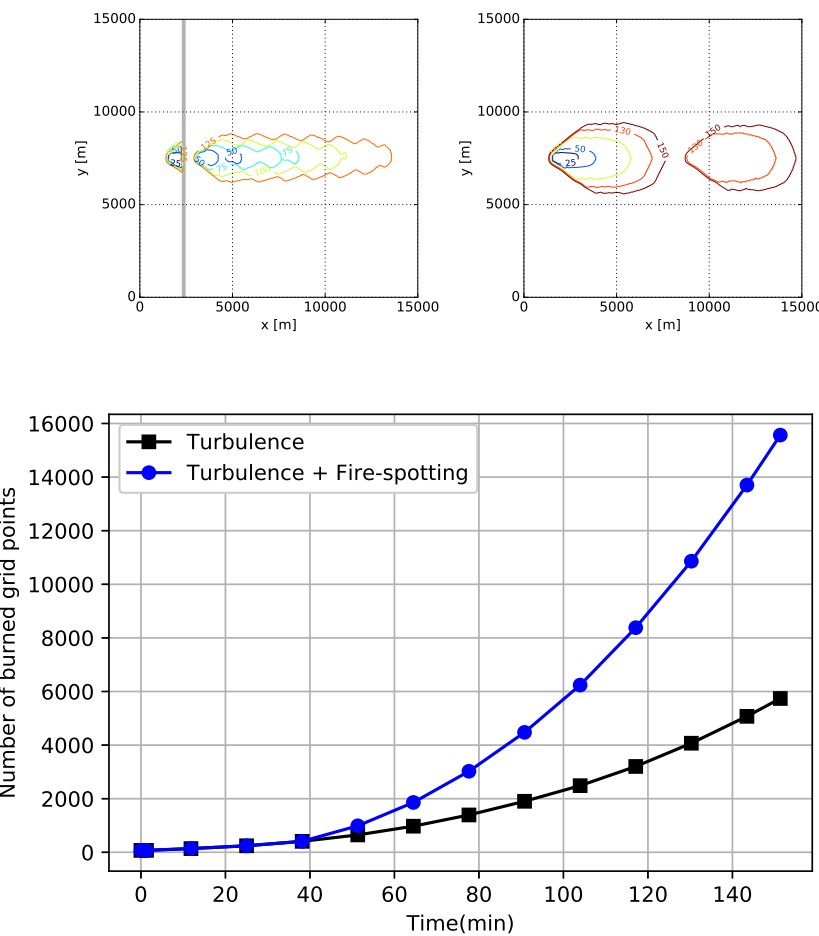

**Figure 4.** Top left panel: Line contours showing the fire perimeter at different time steps with the presence of a fire barrier. The wind velocity is $10\,\mathrm{ms}^{-1}$, fire intensity is $25\,\mathrm{MWm}^{-1}$ and diffusion coefficient is $0.15\,\mathrm{m}^2\mathrm{s}^{-1}$. The $x$ and $y$ axis of the plot are scaled by a factor of 4. The same plot is proposed at the right, but with $U = 20\mathrm{ms}^{-1}$ and no barrier. Bottom panel: A comparison of the total burned area at different time steps when only turbulence is considered (black) and when both turbulence and fire-spotting are included (red). The total burned area is simply the number of burned grid points at any each instant. For both line plots, the wind velocity is $10\,\mathrm{ms}^{-1}$, fire intensity is $25\,\mathrm{MWm}^{-1}$ and diffusion coefficient is $0.15\,\mathrm{m}^2\mathrm{s}^{-1}$.

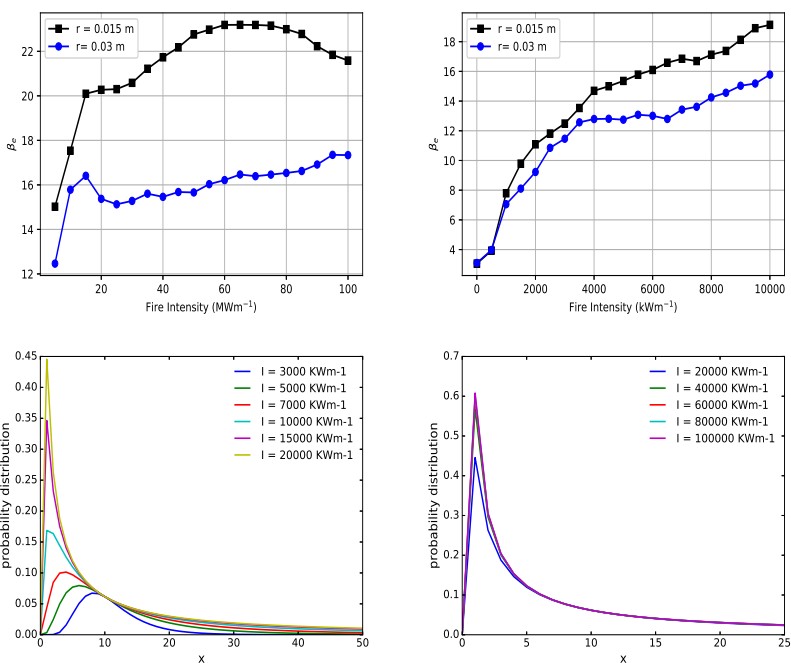

**Figure 5.** Top panel: Line plot showing the sensitivity of the formulation to different values of fire intensity over constant wind conditions ($10\,\mathrm{ms}^{-1}$) and constant firebrand radius ($0.015\,\mathrm{m}$). The sensitivity is measured in terms of the total increase in the burned area when both fire-spotting and turbulence are included over the case when no random effects are considered. The parameter $\beta_e$ is defined as : $\beta_e = (\mathbf{x}_{random} - \mathbf{x}_{no-random})/\mathbf{x}_{no-random}$. The diffusion coefficient $\mathscr{D}$ is $0.15\,\mathrm{m}^2\mathrm{s}^{-1}$.

Bottom panel: The line plots show the lognormal distribution for selected values of fire intensity $I$ but constant values of wind speed $U = 10\,\mathrm{ms}^{-1}$ and firebrand radius $r = 0.03\,\mathrm{m}$. According to the physical parametrisation, the plots can also be interpreted as the behaviour of the lognormal distribution for varying values of parameter $\mu$ and constant value of parameter $\sigma$.

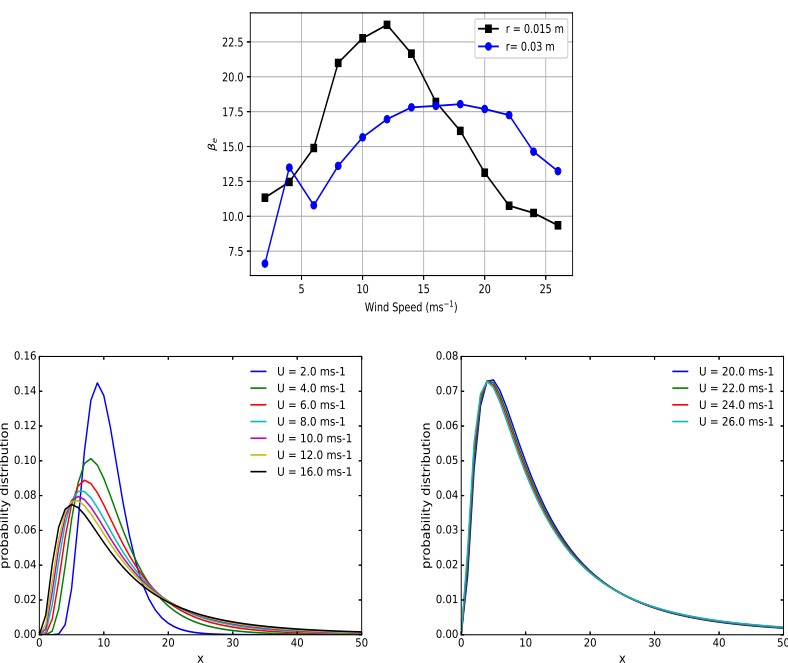

**Figure 6.** Top panel: Line plot showing the sensitivity of the formulation to different wind conditions and radii, when fire intensity is constant ($50\,\mathrm{MWm}^{-1}$). The measure of effective increase in area ($\beta_e$) and other simulation parameters are the same as defined in Figure 5. The adjacent figure shows the lognormal distribution for selected values of wind speed, when fire intensity is $50000\ \mathrm{kWm}^{-1}$ and radius of firebrand is $0.015$ m.

Bottom panel: The line plots show the lognormal distribution for selected values of wind speed $U$ but constant values of fire intensity $I = 5000\,\mathrm{kWm}^{-1}$ and firebrand radius $r = 0.03\,\mathrm{m}$. According to the physical parametrisation, the plots can also be interpreted as the behaviour of the lognormal distribution for varying values of parameter $\sigma$ and constant value of parameter $\mu$.