# Peer review of "RandomFront 2.3 A physical parametrisation of fire-spotting for operational fire spread models: Implementation in WRF-Sfire and response analysis with LSFire+"

_Geoscientific Model Development, 2018_

## Referee Comment (RC1) · Anonymous Referee #1 · 30 Mar 2018

Authors present a physical parametrization of a model developed by one of the authors to include random processes into operational fire spread models as a post-processing scheme. These random processes include mainly fire-spotting, but also turbulence. Authors applied this scheme to wildfire spread models based on the Level Set Method.

The topic of the paper is well suitable for the journal, and of current interest as wildfires are increasing concerns in the research community in the context of climate change.

The organization of the paper is correct. The state of the art included in the introduction is complete and the bibliography used is updated. I suggest revising also the following paper: Calculation of Spotting Particles Maximum Distance in Idealised Forest

[Figure]

Fire Scenarios José C. F. Pereira, José M. C. Pereira, André L. A. Leite, and Duarte M. S. Albuquerque, Journal of Combustion, Volume 2015 (2015), Article ID 513576, http://dx.doi.org/10.1155/2015/513576 In this section, there is a minor typesetting error in line 33 of page 2, "no of the them..."

Section two is a resume of the mathematical model that is more deeply described in previous works of one of the author.

Section three is the main part of the article, where the physical parametrization is detailed. To make it easier to read and understand we suggest including a notation table. Does U represent the meteorological wind?

In section four, a more detailed description of the experiments is required, for example the simulation area size and the computational cost of the experiments. Why the turbulent diffusion coefficient is assumed to be $0.15 m^2 s^{-1}$? Sentence of line 15 in page 8 should be detailed with data and/or references.

Section 5 deserves more attention. We suggest an improvement on figure 1, top panel by adding intermediate contour lines between 25 and 60 min. In this top panel are considered both, turbulence and fire-spotting? The parameter beta_e is an interesting idea to evaluate the effective increase in the burned area but, we found that the sensitivity of the model to the wind speed, fire intensity and firebrand radius is not complete with the experiments developed. A global sensitivity analysis should be performed in order to a comprehensive study of the physical parametrization of the model.

In conclusion section, sentences between line 20 and 25 in page 10 raise doubts. When the wind speed or fire intensity is high, ROS is higher, and the fire front quickly achieves secondary fires, so beta_e could be smaller, but maybe this does not mean that the firebrands fail to cause new ignitions. When is measured beta_e in figure 2?

With the improvements suggested, the paper can be accepted.

---

## Short Comment (SC1) · 4 Apr 2018

Dear authors,

in my role as Executive editor of GMD, I would like to bring to your attention our Editorial version 1.1: http://www.geosci-model-dev.net/8/3487/2015/gmd-8-3487-2015.html This highlights some requirements of papers published in GMD, which is also available on the GMD website in the 'Manuscript Types' section: http://www.geoscientific-model-development.net/submission/manuscript_types.html In particular, please note that for your paper, the following requirement has not been met in the Discussions paper:

[Figure]

- "The main paper must give the model name and version number (or other unique identifier) in the title."

Please provide a name and the version number of your parametrisation in the title of your revised manuscript. Note, that both, name and version number, are important to identify your parametrisation and the version of your parametrisation.

As explained in https://www.geoscientific-model-development.net/about/manuscript_types.html GMD is encouragingauthors to upload the program code of models (including relevant data sets) as supplement or make the code and data of the exact model version described in the paper accessible through a DOI (digital object identifier). In case your institution does not provide the possibility to make electronic data accessible through a DOI you may consider other providers (eg. zenodo.org of CERN) to create a DOI. Please note that in the code accessibility section you can still point the reader to how to obtain the newest version.

Yours, Astrid Kerkweg

---

## Referee Comment (RC2) · Anonymous Referee #2 · 13 Apr 2018

General comments: The paper outlines a physical parameterization of the effects of spotting and atmospheric turbulence on the fire propagation. The main concept utilized in this approach is to treat them as random processes and apply the correction to the level set function used to trace the fire progression. The presented method is an extension of the original approach focussed on the turbulent effects and published by the authors previously. The subject is definitely of a great importance from the fire modeling standpoint and fits the journal well.

Specific comments on the structure of the paper: The paper is generally well written, but its organization should be revised. There are three elements that are missing in the

paper. First is a clear description of the physics of the process being modeled, as well as approximations and assumptions necessary to formulate the model. The second one is a simple description of the input into the model, the output from the model and what the model does in practical terms. In the current form, all the equations need to be deciphered and linked together in order to reconstruct the proposed data flow. There are four distinct processes that should be clearly outlined, and framed by the corresponding input and output parameters: -Firebrand generation -Firebrand lofting -Firebrand horizontal transport and mass loss -Secondary ignitions

The third missing element is a detailed description of the modeling setup used in this. Without that, is not clear how the results presented in the paper have been obtained, and their replication is impossible. Also, the choice of parameter values used in the study should be discussed (e.g. the diffusion coefficient).

Specific issues: The main issue that needs to be addressed is the model formulation in terms of the firebrand lifting. In the proposed approach Wang (2011) formula is used (see eq. 5 on p 5). Unfortunately, this formula is not used as intended. Contrary to what is suggested, it represents the downwind propagation distance as a function of the loftable height being a function of the firebrand size. 'H' in equation 5 has nothing to do with the maximum plume top height provided by the Sofiev scheme. It is a maximum loftable height for a firebrand of radius r, (which is a function of the up-draught flow), not the plume top height which defines at what height the vertical in-plume velocity goes to zero. Using the plume top height in this context is incorrect because firebrands do not behave as air parcels due to their mass and non-zero fall speed. I think this problem may have led to doubtful conclusions presented in section 6.

A logical solution would be to use the resolved velocity profile from the coupled fireatmosphere model to compute the maximum loftable height based on the updraft strength at any given height. Alternatively, a simplified 1D column model (for instance Freitas model 2007, 2010) could be used to compute the vertical profile of the updraft velocity needed to assess the lifting height. The aforementioned issue is related to the choice of the host fire model used for the simulations. The paper could be strengthened by implementing the proposed method into an actual coupled fire-atmosphere model like WRF-SFIRE, or FOREFIRE. There are two reasons for that. One is the practical aspect of the presented work, and the other one is the physical representation of the spotting process itself. The coupled fire-atmosphere models offer many advantages over uncoupled models especially important from the standpoint of the fire-spotting modeling. They resolve convective updrafts uplifting firebrands, they render the flow downwind from the fire transporting the firebrands, and they account for the interactions between the secondary spot fires and the main fire front. All these aspects are missing in the actual analysis presented in the paper, but when added they would significantly improve the scientific content of this paper as well as its broader impact. The WRF-SFIRE mentioned by the authors multiple times, or any other community open source fire model would be a great choice for a host-model assuring broader application of the presented method, reaching beyond theoretical discussions.

I would also suggest reorganizing the discussion of the model limitation following the general comments. For instance how the vertical wind shear is taken into account? How the wind modification by the fire itself is represented, do the secondary ignitions interact with the main fire front etc.

Adding a table with the description of all the symbols used in this study would be very helpful. Its lack makes following the model description very difficult.

---

## Author Comment (AC1) · 17 Aug 2018

Dear Referees and Dr. A. Kerkweg,

we thank you very much for carefully reading of our manuscript and your support to our work through your suggestions for its improvement.

Actually, by collecting and following your comments, we can definitely say that the perspectives of our research are now much more focused on the fact that when dealing with applied problems the applied aims should always be the priority. We recognise that before the peer-review process our research was still in an "academic approach" with

the concrete risk of making eâĘŢorts with no interest to the academic community; as we focused on an applied problem, and with no interest to an applied community. For this reason, we acknowledge your positive and constructive actitude directed to find the potentialities of our work and pushing us to put them into a useful framework.

This change of perspective took us one month and an half more than the standard deadline for re-submission. Hence we really thank also the editor Kato Tomomichi and the editorial support Anna Feist-Polner for allowing and managing such extension.

This change of perspective required also a change in the title, where the names of the considered models are now explicit, and the work by two further co-authors. In particular the first author is changed. In the section "Author contributions" we provide the role of all authors.

In general, the manuscript has been strongly revised and the implementation of our routines for including fire-spotting in the large-scale operational code WRF-Sfire (as required by Referee 2) embodies the main change. We did not include the global sensitivity analysis required by Referee 1, mainly because it was a long work (6 months) done in collaboration with a group at CERFACS, Toulouse, France, and its presentation and discussion needs many pages. Hence, we think that such analysis deserves a separate paper. However, this separate paper is next to submission and we intend to upload it in arxiv.org before the end of August 2018.

In the attached document we include the rebuttal letter for the Referee and, because of the many changes, the PDF files of the revised version with and without marked changes.

Sincerely Yours,

The authors

Please also note the supplement to this comment:

https://www.geosci-model-dev-discuss.net/gmd-2018-33/gmd-2018-33-AC1-supplement.zip

---

## Author Comment (AC4) · 17 Sep 2018

The referee asked for a sensitivity analysis of model. We performed such analysis as a separate paper and now submitted elsewhere. Here is the link to the arxiv version:

https://arxiv.org/pdf/1809.05430.pdf

Best regards, The authors
arXiv:1809.05430v1 [physics.ao-ph] 14 Sep 2018

Surrogate-based global sensitivity analysis for turbulence and fire-spotting effects in regional-scale wildland fire modeling

A. Trucchiaa,b,\*, V. Egorovaa, G. Pagninia,c, M. C. Rochouxd,\*\*

\*BCAM - Basque Center for Applied Mathematics, Alameda de Maxerdo 14, E-\$1690 Biblos, Basque Country, Spain bUniversity of the Basque Country, UPV/EHIV, Barrio Sarriena 3/n, 4890 Leioa, Basque Country, Spain "Rerbasque - Basque Foundation for Science, Calle de María Daía de Haros 3, E49013 Biblos, Basque Country, Spain "CECI, University of Toulouse, CNRS, CERFACS, 24 Serveu Gasquel Corolas, 51057 Toulouse cede 1, France

**Abstract**

In presence of strong winds, wildfires feature nonlinear behavior, possibly inducing fire-spotting. We present a global sensitivity analysis of a new submodel for turbulence and fire-spotting included in a wildfire spread model based on a stochastic representation of the fireline. To limit the number of model evaluations, fast surrogate models based on generalized Polynomial Chaos (gPC) and Gaussian Process are used to identify the key parameters affecting topology and size of burnt area. This study investigates the application of these surrogates to compute Sobol' sensitivity indices in an idealized test case. The wind is known to drive the fire propagation. The results show that it is a more general leading factor that governs the generation of secondary fires. This study also compares the performance of the surrogates for varying size and type of training sets as well as for varying parameterization and choice of algorithms. The best performance was achieved using a gPC strategy based on a sparse least-angle regression (LAR) and a lowdiscrepancy Halton's sequence. Still, the LAR-based gPC surrogate tends to filter out the information coming from parameters with large length-scale,

\*Email: atrucchia@bcamath.org

\*\*Email: melanie.rochoux@cerfacs.fr

Preprint submitted to XXX

September 17, 2018

---

## Author Response (AR1)

**Rebuttal Letter to Anonymous Referee #1**

In the present Letter, we report in *italic* the report by the Referee and in Times New Roman our reply.

Authors present a physical parametrization of a model developed by one of the authors to include random processes into operational fire spread models as a post-processing scheme. These random processes include mainly firespotting, but also turbulence. Authors applied this scheme to wildfire spread models based on the Level Set Method. The topic of the paper is well suitable for the journal, and of current interest as wildfires are increasing concerns in the research community in the context of climate change. The organization of the paper is correct. The state of the art included in the introduction is complete and the bibliography used is updated. I suggest revising also the following paper: Calculation of Spotting Particles Maximum Distance in Idealised Forest Fire Scenarios José C. F. Pereira, José M. C. Pereira, André L. A. Leite, and Duarte M. S. Albuquerque, Journal of Combustion, Volume 2015 (2015), Article ID 513576, http://dx.doi.org/10.1155/2015/513576 In this section, there is a minor typesetting error in line 33 of page 2, no of the them...

We have included the suggested reference in the introduction where other approaches are reviewed. In particular, where approaches based on LES are reported. The typo has been corrected.

Section two is a resume of the mathematical model that is more deeply described in previous works of one of the author.

Section three is the main part of the article, where the physical parametrization is detailed. To make it easier to read and understand we suggest including a notation table. Does U represent the meteorological wind?

In the revised version we have included a table with symbols.

In section four, a more detailed description of the experiments is required, for example the simulation area size and the computational cost of the experiments.

The required information on the simulation set-up are included, and the computational costs are reported in the section "Code availability".

Why the turbulent diffusion coefficient is assumed to be 0.15m2 s-1? Sentence of line 15 in page 8 should be detailed with data and/or references.

At the end of section 3, we have included a more detailed estimation of the value of the diffusion coefficient.

Section 5 deserves more attention. We suggest an improvement on figure 1,

top panel by adding intermediate contour lines between 25 and 60 min.

We have included intermediate contour lines.

In this top panel are considered both, turbulence and fire-spotting? The parameter beta\_e is an interesting idea to evaluate the effective increase in the burned area but, we found that the sensitivity of the model to the wind speed, fire intensity and firebrand radius is not complete with the experiments developed. A global sensitivity analysis should be performed in order to a comprehensive study of the physical parametrization of the model.

We did not include a general sensitivity analysis as required by the Referee. The main reason is that such work is the subject of an other paper in preparation and next to submission. We intend to upload it in arxiv.org before the end of August 2018. Actually, the sensitivity analysis to input parameters and the uncertainty quantification on outputs were performed by A. Trucchia during a 6-months periods at CERFACS, Toulouse, France, in collaboration with M. Rochoux. The description of the adopted methods, comprehensive list of figures and the discussion of the plots needed many pages, hence we think that such analysis deserves a separate paper.

In conclusion section, sentences between line 20 and 25 in page 10 raise doubts. When the wind speed or fire intensity is high, ROS is higher, and the fire front quickly achieves secondary fires, so beta\_e could be smaller, but maybe this does not mean that the firebrands fail to cause new ignitions. When is measured beta\_e in figure 2?

The referee is right. We changed that explanation.

With the improvements suggested, the paper can be accepted.

We hope the Referee considers the revised version properly improved and deserving publication.

**Rebuttal Letter to Anonymous Referee #2**

In the present Letter, we report in *italic* the report by the Referee and in Times New Roman our reply.

General comments: The paper outlines a physical parameterization of the effects of spotting and atmospheric turbulence on the fire propagation. The main concept utilized in this approach is to treat them as random processes and apply the correction to the level set function used to trace the fire progression. The presented method is an extension of the original approach focussed on the turbulent effects and published by the authors previously. The subject is definitely of a great importance from the fire modeling standpoint and fits the journal well.

Specific comments on the structure of the paper: The paper is generally well written, but its organization should be revised. There are three elements that are missing in the paper.

First is a clear description of the physics of the process being modeled, as well as approximations and assumptions necessary to formulate the model.

The physical idea and approximations are written in section 2, for what concerns the fundamentals, and they are reported at the beginning of section 3, for what concerns the application.

The second one is a simple description of the input into the model, the output from the model and what the model does in practical terms. In the current form, all the equations need to be deciphered and linked together in order to reconstruct the proposed data flow.

The flow of the model execution is reported now in section 4.

There are four distinct processes that should be clearly outlined, and framed by the corresponding input and output parameters: -Firebrand generation -Firebrand lofting -Firebrand horizontal transport and mass loss -Secondary ignitions.

The description of these points is now included in section 3.

The third missing element is a detailed description of the modeling setup used in this. Without that, is not clear how the results presented in the paper have been obtained, and their replication is impossible.

More information on simulation set-up are included. Moreover, the section "Code availability" has been re-written and includes further information.

Also, the choice of parameter values used in the study should be discussed (e.g. the diffusion coefficient).

The estimation of the diffusion coefficient is now explained in more detail at the end of section 3.

Specific issues: The main issue that needs to be addressed is the model formulation in terms of the firebrand lifting. In the proposed approach Wang (2011) formula is used (see eq. 5 on p 5). Unfortunately, this formula is not used as intended. Contrary to what is suggested, it represents the downwind propagation distance as a function of the loftable height being a function of the firebrand size. H in equation 5 has nothing to do with the maximum plume top height provided by the Sofiev scheme. It is a maximum loftable height for a firebrand of radius r, (which is a function of the up-draught flow), not the plume top height which defines at what height the vertical in-plume velocity goes to zero. Using the plume top height in this context is incorrect because firebrands do not behave as air parcels due to their mass and non-zero fall speed. I think this problem may have led to doubtful conclusions presented in section 6. A logical solution would be to use the resolved velocity profile from the coupled fire-atmosphere model to compute the maximum loftable height based on the updraft strength at any given height. Alternatively, a simplified 1D column model (for instance Freitas model 2007, 2010) could be used to compute the vertical profile of the updraft velocity needed to assess the lifting height.

The referee is right, and we are really grateful for highligting this error. In the revised version we decided to remain within the framework mainly provided by Wang (2011), because other parts of the parametrisation also followed this framework. This error is now corrected in the description (see section 3) and in the simulations. Conclusions are also revised.

The aforementioned issue is related to the choice of the host fire model used for the simulations. The paper could be strengthened by implementing the proposed method into an actual coupled fire-atmosphere model like WRF-SFIRE, or FOREFIRE. There are two reasons for that. One is the practical aspect of the presented work, and the other one is the physical representation of the spotting process itself. The coupled fire-atmosphere models offer many advantages over uncoupled models especially important from the standpoint of the fire-spotting modeling. They resolve convective up-drafts uplifting firebrands, they render the flow downwind from the fire transporting the firebrands, and they account for the interactions between the secondary spot fires and the main fire front. All these aspects are missing in the actual analysis presented in the paper, but when added they would significantly improve the scientific content of this paper as well as its broader impact. The WRF-SFIRE mentioned by the authors multiple times, or any other community open source fire model would be a great choice for a host-model assuring broader application of the presented method, reaching beyond theoretical discussions. I would also suggest reorganizing the discussion of the

model limitation following the general comments. For instance how the vertical wind shear is taken into account? How the wind modification by the fire itself is represented, do the secondary ignitions interact with the main fire front etc.

The proposed parametrisation is now implemented in WRF-Sfire. A simple test case is studied and discussed.

Adding a table with the description of all the symbols used in this study would be very helpful. Its lack makes following the model description very difficult.

A table of symbols has been included.

We hope the Referee considers the revised version properly improved and deserving publication.

**Reply to Executive Editor of GMD A. Kerkweg**

**Dear authors,**

in my role as Executive editor of GMD, I would like to bring to your attention our Editorial version 1.1: http://www.geosci-model-dev.net/8/3487/2015/gmd-8-3487-2015.html This highlights some requirements of papers published in GMD, which is also available on the GMD website in the Manuscript Types section: http://www.geoscientific-model-development.net/submission/manuscript\_types.html In particular, please note that for your paper, the following requirement has not been met in the Discussions paper:

"The main paper must give the model name and version number (or other unique identifier) in the title."

Please provide a name and the version number of your parametrisation in the title of your revised manuscript. Note, that both, name and version number, are important to identify your parametrisation and the version of your parametrisation. As explained in https://www.geoscientific-modeldevelopment.net/about/manuscript\_types.html GMD is encouragingauthors to upload the program code of models (including relevant data sets) as supplement or make the code and data of the exact model version described in the paper accessible through a DOI (digital object identifier). In case your institution does not provide the possibility to make electronic data accessible through a DOI you may consider other providers (eg. zenodo.org of CERN) to create a DOI. Please note that in the code accessibility section you can still point the reader to how to obtain the newest version.

**Yours, Astrid Kerkweg**

We thank the Executive Editor for remarking this point that pushed us towards a more pragmatic approach. Actually, before the submission we did not know how to proceed with respect to this point, as we understood our work as a proof-of-concept. Now, we have numbered all the versions of the parametrisation, provided their chronology, and named all the codes we have used.

In this respect, at the end of the revised version we have included in the *Code availability* section the information concerning the used codes and we report the link to the official git repository of BCAM where the both the implementations of RandomFront 2.3 used in this paper are freely available, namely at the address:

[revised manuscript text omitted]

$$\underline{\mu\sigma} = \underline{\mathcal{H}} \underbrace{\frac{1}{2z_p}}_{\sim \sim \sim \sim} \ln \left( \underbrace{\frac{3\rho_a C_d}{2\rho_f} \frac{\mathcal{U}^2}{rg}}_{\sim \sim \sim} \right) \frac{1/2}{r},\tag{10}$$

$$\quad \underline{\sigma\mu} = \frac{1}{2z_p} \ln \mathcal{H} \left( \frac{U^2}{\underline{rg}} \frac{3}{2} \frac{\rho_a}{\rho_f} C_d \right) \overset{1/2}{\sim} . \tag{11}$$

In this parametrisation of the fire-spotting, the quantification

10

We chose such parametrization of  $\mu$  and  $\sigma$  is chosen to provide a most rational description for the transport of firebrands. The parameter in order to de-lineate the governing parameters for lofting and transport mechanisms respectively. We hypothesise that the definition of  $\mu$  is parametrised to characterize the lofting covers the essential input parameters needed to describe the

25 lofting mechanism of the firebrands inside the convective column. The relative density  $\rho_a/\rho_f$  and atmospheric drag quantify the buoyant forces experienced by the firebrand; hence it is appropriate to include these quantities in the definition of  $\mu$ to describe the maximum allowable height for each firebrand in varying fire intensities. The density ratio  $\rho_a/\rho_f$  also limits the maximum allowable height for each firebrand. Substituting maximum loftable height from Eq. (8) in  $\mu$  gives:

$$\mu = 3.52 \times 10^5 \left(\frac{\rho_a}{\rho_f} C_d\right)^2 \left(\frac{I}{H_c}\right)^{5/3} r^{-3/2} g^{-5/2}.$$
(12)

- 30 The radius of the firebrand r and the fuel density are important ingredients to determine the height of the lofted firebrands. On the other hand,  $\sigma$  is parametrised hypothesized to define the transport of the firebrands under the effect of the wind after they are ejected horizontal wind after ejection from the convective column. In a wind driven regime of fire-spotting, the flight path of the firebrand is affected by its size and firebrands beyond a critical size cannot be steered by the prevailing wind. This critical size is defined as the maximum liftable radius  $r_{max} = U^2/g$ . It is interesting to note that the dimensionless ratio  $U^2/(rg)$  is
- 5 also known as The definition of  $\sigma$  includes a dimensionless ratio  $\mathcal{F} = \mathcal{U}^2/(rg)$  which is analogous to the Froude numberand, , which quantifies the balance between the inertial and the inertial and gravitational forces experienced by the firebrand. All firebrands with  $r \leq \mathcal{U}^2/g$  can be transported by the horizontal wind.

In this model, the phenomenon of fire-spotting is assumed to occur together with the turbulent heat flux around the fire, and the turbulent diffusion coefficient  $\mathscr{D}$  is utilised as a measure of the turbulent heat transfer generated by the fire. It is parametrised

10 in terms of the Nusselt number Nu. Nusselt number defines the ratio between the convective and conductive heat transfer in fluids and is defined as

**$Nu = (\mathscr{D} + \chi)/\chi\,,$**

15

where  $\chi Nu = (\mathscr{Q} + \chi)/\chi$  where  $\chi = 2 \times 10^{-5} \text{ m}^2 \text{s}^{-1}$  is the thermal diffusivity of air at ambient temperature. Experimentally, it is shown that Nusselt number is related to Rayleigh number as  $Nu \simeq 0.1 Ra^{1/3}$  (Niemela and Sreenivasan, 2006). Rayleigh number is defined as  $Ra = \gamma \Delta T g h^3/(\nu \chi)$ , where  $\gamma = 3.4 \times 10^{-3} \text{ K}^{-1}$  is the thermal expansion coefficient, h is the dimensional diffusivity of a statement of the temperature.

[revised manuscript text omitted]

$$\underline{\underline{V}}_{\underline{\underline{V}}_{\underline{\underline{r}}\underline{o}\underline{s}}}(\underline{\underline{x}}\underline{x},t) = \frac{I(1+f_w)}{H\,\alpha\,\omega_0},\tag{14}$$

where *I* is the fire intensity,  $H = 22000 \text{ KJKg}^{-1}$ ,  $H = 22000 \text{ kJkg}^{-1}$  is the fuel low heat of combustion,  $\omega_0 = 2.243 \text{ Kgm}^{-2}$  $\omega_0 = 2.243 \text{ kgm}^{-2}$  is the oven-dry mass of the fuel and the functional dependence on the wind is included through the factor  $f_w$ . The user has flexibility to introduce a different ecosystem in the simulations by modifying the the parameters H and  $\omega_0$ . The parameter  $\alpha$  is chosen to guarantee that the maximum ROS is always equal to the ROS prescribed by the Byram formulation.

Sensitivity The response of the formulation to depict the different firebrand landing distributions is highlighted through two sets of test cases. In the first test case, the wind conditions and the size of the firebrands are assumed to be constant as the fire intensity changes. In the second test case, the fire intensity is assumed to be non-changing constant and the simulations for different wind conditions are carried out. The second test case is also repeated for a different radius of the firebrand different firebrand radii

10 firebrand radii.

[revised manuscript text omitted]

The simulated domain for the response analysis is chosen as a rectangle of dimensions [0m, 6000m] × [0m, 6000m]. The
simulations start at time t = 0 min and end at time t = 140 min. The grid spacing is Δx = Δy = 20m. At time t = 0 min the initial fire-line is a circle of radius 180 m centered at xc = (720m, 3000m). The horizontal wind has been assumed in this simulation set-up as a constant field parallel to the vector j = (1,0) and with modulus |Uh| = |(U,V)|.

**Response analysis to fire intensity**

An increase in the fire intensity of the wildfire increases. Constant wind velocity causes an increase in the burned area (see

30 Eqn (14) for the definition of the ROS); at the same time, the fire-spotting behaviour is also affected by any change in *I*. The parameter  $\beta_e$  allows us to identify the contribution of fire-spotting towards the fire-propagation. The top panel of Fig. 5 shows

the change in the burned area under the combined effect of turbulence and fire-spotting with increase in the fire-intensity. The two line plots correspond to a constant wind speed  $(10 \text{ ms}^{-1})$  and the firebrand radius (but two firebrand radii, i.e., 0.015 m) are considered for these simulations. From and 0.030 
[revised manuscript text omitted]

| $\tau$ , ignition delay of firebrands                                            | $\gtrsim$                           |
| I, fire-line intensity                                                           | $MWm^{-1}$                          |
| Ugas, Vertical gas flow                                                          | $ms^{-1}$                           |
| r, radius of spherical firebrand                                                 | m                                   |
| rmax, maximum loftable radius for spherical firebrand                            | m                                   |
| $\mathcal{H}$ , maximum loftable height for spherical firebrands                 | m                                   |
| Vrest rate of spread                                                             | $ms^{-1}$                           |

| Quantity                         | Unit of measurement                           | First Test Case  | Second Test Case                                 |
|----------------------------------|-----------------------------------------------|------------------|--------------------------------------------------|
| S
D                           | $\underline{m^2 s^{-1}}$                      | 0.15             | 0.15                                             |
| $\underset{\sim}{\underline{U}}$ | $ms^{-1}$                                     | $10_{\sim}$      | $2 \div 26$                                      |
| $\stackrel{I}{\sim}$             | $\underset{\text{WWm}^{-1}}{\text{MWm}^{-1}}$ | $5 \div 100$     | 50                                               |
| $\stackrel{r}{\sim}$             | $\stackrel{\mathrm{m}}{\sim}$                 | 0.015            | $\underbrace{0.015 \div 0.03}_{0.015 \div 0.03}$ |
| $\stackrel{\mathcal{T}}{\sim}$   | $\stackrel{\mathrm{s}}{\sim}$                 | $\frac{1}{\sim}$ | 1                                                |

Table 2. Values of the main parameters for numerical simulations performed with LSFire+.

---

## Referee Report (RR1)

**gmd-2018-33 Submitted on 06 Feb 2018**

RandomFront 2.3 A physical parametrisation of fire-spotting for operational fire spread models: Implementation in WRF-Sfire and response analysis with LSFire+ Andrea Trucchia, Vera Egorova, Anton Butenko, Inderpreet Kaur, and Gianni Pagnini

This revised version of the previous paper **Physical parametrisation of fire-spotting for operational fire spread models: response analysis with a model based on the Level Set Method** has important improvements that I value very positively, especially regarding the examples, coupling the RandomFront post-processing scheme with two fire spread models, one based on the level set method, and the other coupled with an atmosphere model.

As in my previous review, again I think that the topic of the paper is well suitable for the journal, and of current interest as wildfires are increasing concerns in the research community in the context of climate change and the new paradigm of wildfires that climate change has causing: sixth generation wildfires.

The organization of the paper is correct. The state of the art included in the introduction is complete and the bibliography used is updated. The improvements in the article are very positive, especially in the examples.

I suggest accepting the publication, with two minors changes:

1- Pag 17, par 20-26: It would be interesting to explain what part of the computational cost corresponds to the post-processing routine. The computational cost of the both models (WRF-Sfire and LSFire++) is detailed, but it is not clear if this includes the RandomFront computational cost or not.

2. Figures 1-3 captions should include an explanation about the meaning of dashed lines, this is explained in the text, but I suggest adding it in the captions too.

---

## Author Response (AR2)

Dear Referee,

we thank you very much for your second revision and final suggestions for improving the manuscript.

In particular, concerning remark 1

*1. Pag 17, par 20-26: It would be interesting to explain what part of the computational cost corresponds to the post-processing routine. The computational cost of the both models (WRF-Sfire and LSFire+) is detailed, but it is not clear if this includes the RandomFront computational cost or not.*

we have included the following lines:

The 80% of the computational cost in both cases, i.e., `WRF-Sfire` and `LSFire+`, is due to the post-processing routine `RandomFront 2.3`. This computational time can be reduced in the future through a further code optimisation.

and concerning remark 2

*2. Figures 1-3 captions should include an explanation about the meaning of dashed lines, this is explained in the text, but I suggest adding it in the captions too.*

we have included in the captions of Figures 1-3 the following line:

Firefront is reported by a dashed line.

Thank you very much.
Sincerely Yours,
The authors

**List of Changes**

1. At page 17, at the end of the section "Code availability" we have included the following lines:

*The 80% of the computational cost in both cases, i.e.,* `WRF-Sfire` *and* `LSFire+`, *is due to the post-processing routine* `RandomFront 2.3`. *This computational time can be reduced in the future through a further code optimisation.*

2. In the captions of Figures 1-3 we have included the following line:

*Firefront is reported by a dashed line.*

[revised manuscript text omitted]